# Stratified Kelvin-Helmholtz turbulence of compressible shear flows

Romit Maulik [1] and Omer San [1]

[1]School of Mechanical and Aerospace Engineering, Oklahoma State University, Stillwater, Oklahoma 74078

*Correspondence to:* Omer San (osan@okstate.edu)

**Abstract.** We study scaling laws of stratified shear flows by performing high-resolution numerical simulations of inviscid compressible turbulence induced by Kelvin-Helmholtz instability. An implicit large eddy simulation approach is adapted to solve our conservation laws for both two-dimensional (with a spatial resolution of $16,384^2$) and three-dimensional (with a spatial resolution of $512^3$) configurations utilizing different compressibility characteristics such as shocks. For three-dimensional turbulence, we find that both the kinetic energy and density-weighted energy spectra follow the classical Kolmogorov $k^{-5/3}$ inertial scaling. This phenomenon is observed due to the fact that the power density spectrum of three-dimensional turbulence yields the same $k^{-5/3}$ scaling. However, we demonstrate that there is a significant difference between these two spectra in two-dimensional turbulence since the power density spectrum yields a $k^{-5/3}$ scaling. This difference may be assumed to be a reason for the $k^{-7/3}$ scaling observed in the two-dimensional density-weight kinetic every spectra for high compressibility as compared to the $k^{-3}$ scaling traditionally assumed with incompressible flows. Further inquiries are made to validate the statistical behavior of the various configurations studied through the use of the Helmholtz decomposition of both the kinetic velocity and density-weighted velocity fields. We observe that the scaling results are invariant with respect to the compressibility parameter when the density-weighted definition is used. Our two-dimensional results also confirm that a large inertial range of the solenoidal component with the $k^{-3}$ scaling can be obtained when we simulate with a lower compressibility parameter, however, the compressive spectrum converges to $k^{-2}$ for a larger compressibility parameter.

## 1 Introduction

Turbulence is a highly nonlinear multiscale phenomenon which is ubiquitous in nature. It poses some of the most challenging problems in classical physics as well as in computational mathematics. Understanding the nature of compressible turbulence is of paramount importance. Highly compressible turbulence plays an important role in star formation control in dense molecular clouds (Padoan and Nordlund, 2002; Mac Low and Klessen, 2004; Mac Low et al., 1998) and are responsible for important design considerations in many engineering applications. Therefore, there have been several investigations into its statistical behavior. Kida and Orszag (1990) studied the mechanics of energy transfer and distribution as well as an examination of small-scale spectra in compressible turbulence with root mean square Mach numbers upto 0.9. Theoretical laws have also been advanced for the statistical behavior of turbulence quantities under the influence of compressibility effects (Shivamoggi, 1992; Lele, 1994; Shivamoggi, 2011; Wang et al., 2013). Kritsuk et al. (2007) utilized an adaptive mesh refinement (AMR) algorithm along with a piecewise parabolic approach for numerical dissipation to obtain scaling tendencies at high Mach number values

for both kinetic energy, density weighted kinetic energy and density power spectra. In addition, structure functions of different orders were also studied and compared to the limiting case of incompressibility. Aluie (2013) provided a theoretical justification of the presence of an inertial scale which is devoid of any effects of molecular viscosity for supersonic turbulence similar to the classical Richardson-Kolmogorov cascade in homogeneous isotropic incompressible turbulence (Kolmogorov, 1941; Vassilicos, 2015). Magnetic effects on the statistical behavior of supersonic turbulence have also been studied keenly due to implications for astrophysical processes such as in Banerjee and Galtier (2013) where two-point correlation function relations were studied.

Scaling laws incorporating magnetic effects in hydrodynamic turbulence have also been proposed, for instance in Iroshnikov-Kraichnan theory (Iroshnikov, 1964; Kraichnan, 1965), where arguments similar to those used in Kolmogorov theory are used to explain statistical properties of small-scale components in velocity and magnetic fields. Extensions to account for the rather tenuous assumption of isotropy in compressible magnetohydrodynamics (MHD) have also been studied by Goldreich and Sridhar (1997). A generalization of the Iroshnikov-Kraichnan and Goldreich-Sridhar spectra to compressible magnetohydrodynamics has been presented by Shivamoggi (2008) where it is also shown to merge with the MHD shockwave spectrum in the limit of infinite compressibility (Kadomtsev and Petviashvili, 1973). A recent review which examines both hydrodynamic and magnetohydrodynamic implementations of supersonic compressible turbulence on statistical quantities can be found in Falceta-Goncalves et al. (2014). In this work, we follow the vast majority of investigations (Shivamoggi, 1992; Ottaviani, 1992; Domaradzki and Carati, 2007; Falkovich et al., 2010; Kuznetsov and Sereshchenko, 2015; Shivamoggi, 2015; Sun, 2016; Westernacher-Schneider et al., 2015; Qiu et al., 2016; Bershadskii, 2016; Sun, 2017; Westernacher-Schneider and Lehner, 2017) by utilizing the phenomenological description of turbulence in Fourier space as well as the utilization of two-point velocity structure functions for the statistical examination of our high fidelity numerical simulations. One of our goals is to investigate scaling laws using a computational framework with moderately high resolutions. We note that several modified energy spectra and anisotropic behaviors have been recently discussed within the context of the Rayleigh–Taylor and Richtmyer-Meshkov instability induced flows (Zhou, 2017a, b). In terms of reference scaling behavior, we shall be comparing our numerical results of the stratified shear layer turbulence simulations against the theories under the assumption of isentropic flow by solving the Euler equations triggered by stratified shear layers in a periodic box domain.

In this work, we shall examine the stratified compressible turbulence that emerges from a classical Kelvin–Helmholtz instability (KHI) formulation. Similar problems have been studied extensively for their incompressible versions (Hopfinger, 1987; Werne and Fritts, 1999; Peltier and Caulfield, 2003; Boffetta and Mazzino, 2017). In this work, both two- and three-dimensional versions of stratification will be examined for their effects on scaling. It must be noted here that two-dimensional turbulence may be assumed to be an appropriate framework for many geophysical applications which exhibit extremely high aspect ratios and indeed, incompressible two-dimensional turbulence forms the cornerstone of geostrophic turbulence theory (Boffetta and Ecke, 2012; Shivamoggi, 1998). Astrophysical considerations have also been explored in Biskamp and Schwarz (2001) where the effects of a magnetohydrodynamic coupling have also been examined on scaling behavior. Our focus shall primarily rest on a comparison of numerically obtained behavior of the density power spectrum, the averaged kinetic energy spectrum and the density weighted kinetic energy spectrum along with second and third order velocity structure functions with their theoret-

ical predictions. Some reference scaling laws (in the incompressible limit) we shall be using for comparison are the classical Kolmogorov scaling (Kolmogorov, 1941) for isotropic three-dimensional (3D) turbulence and Kraichnan scaling (Kraichnan, 1967) for two-dimensional (2D) isotropic turbulence.

A common strategy for the numerical examination of the statistics of highly compressible turbulence is the use of the Eulerian hydrodynamic conservation laws implemented through an implicit large eddy simulation (ILES) methodology (Passot et al., 1988; Blaisdell et al., 1993). This is because it is commonly accepted that an ILES formulation of the Euler equations provides a good estimation for the Navier–Stokes equations in the limit of infinite Reynolds numbers (Bos and Bertoglio, 2006; Zhou et al., 2014; Sytine et al., 2000). However, two conditions must be enforced in order to satisfy the aforementioned assumption. Firstly, vorticity must be introduced via either boundary and/or initial conditions since the Euler equations are incapable of generating vorticity from irrotational flows. Secondly, an artificial viscosity must be incorporated into the simulation mechanism to mimic the preservation of dissipative behavior of the Navier–Stokes equations in the inviscid limit (Moura et al., 2017). The ILES mechanism is a suitable approach for artificial dissipation through the use of numerical truncation errors and is our simulation algorithm of choice for the high fidelity numerical experiments in this investigation.

The question we attempt to address through this work is related to the difference between purely averaged kinetic energy spectra scaling and density weighted spectra scaling for both two- and three-dimensional compressible turbulence. Our observations suggest a different 'packaging' of density in the spectral space for the two-dimensional turbulence case. This is proven conclusively by comparing the differences in density power spectrum behavior for both two- and three-dimensional configurations. It is proposed that the density power spectrum (or in other words the packaging of density at different wavenumbers) may be a reason that causes a variation in the $k^{-3}$ scaling of the density-weighted kinetic energy cascade with changing compressibility (higher compressibilities are observed to show $k^{-7/3}$ scaling) for two-dimensional turbulence as against the constant $k^{-5/3}$ cascade in three-dimensional turbulence. Our results are also validated through the use of the second order structure function behavior with varying compressibility. High fidelity simulation data are generated by utilizing $512^3$ and $16384^2$ degrees of freedom for the three- and two-dimensional cases respectively. We demonstrate that there is no difference in energy spectrum scalings between kinematic and density weighted velocities in three-dimensional simulations since both the power density and velocity spectra scale with the $k^{-5/3}$ scaling. However, we have demonstrated that the difference becomes pronounced in two-dimensional simulations because the power density spectrum scales with $k^{-5/3}$, which is different than the scaling of the kinetic energy spectrum. Furthermore, we have decomposed both the kinetic velocity and density-weighted velocity fields into compressive (curl-free) and solenoidal (divergence-free) components in order to study the effects of compressibility in our two- and three-dimensional setups. Ultimately, it is our aim to link these analyses to nonlinear processes exhibiting very high aspect ratios for astrophysical, heliophysical and plasma physics applications.

## 2 Compressible turbulence

The governing laws utilized for our numerical experiments are given by the Euler equations which may be expressed in their dimensionless differential form as

$$\frac{\partial \rho}{\partial t} + \nabla.(\rho \boldsymbol{u}) = 0 \tag{1}$$

$$\frac{\partial(\rho \boldsymbol{u})}{\partial t} + \nabla.(\rho \boldsymbol{u} \otimes \boldsymbol{u} + p\boldsymbol{I}) = 0 \tag{2}$$

$$\frac{\partial(\rho E)}{\partial t} + \nabla.(\rho E \boldsymbol{u} + p\boldsymbol{u}) = 0 \tag{3}$$

where $\rho$ is the fluid density, $\boldsymbol{u} = \{u, v\}^T \in \mathbb{R}^2$ and $\boldsymbol{u} = \{u, v, w\}^T \in \mathbb{R}^3$ is the flow velocity in a Cartesian co-ordinate system,

$p$ is the static pressure, and $E$ is the total energy per unit mass. Assuming a perfect gas with a ratio of specific heats $\gamma$, the pressure can be determined by an equation of state which closes our coupled governing equations given by

$$p = \rho(\gamma - 1)\left(E - \frac{1}{2}(\boldsymbol{u}.\boldsymbol{u})\right) \tag{4}$$

where we have set $\gamma = 7/5$ in our study. Note that the assumption of the classical equation of state for relating the pressure and total energy of the flow ensures the interaction of solely acoustic and vortical modes (Shivamoggi, 1992). Our computational

domain also exhibits periodic boundary conditions in all directions.

### 2.1 Stratified Kelvin-Helmholtz instability

The stratified Kelvin-Helmholtz instability (KHI) test-case is a famous problem which manifests itself when there is a velocity difference at the interface between two fluids of different densities (Thomson, 1871). It can commonly be observed through experimental observation, numerical simulation and it is also visible in many natural phenomena such as for example in situa-

tions with wind flow over bodies of water causing wave formation and in the planet Jupiter's atmosphere between atmospheric bands moving at different speeds (Hwang et al., 2012). The study of this instability in a benchmark formulation reveals key information about the transition to turbulence for two fluids moving at different speeds. For these practical applications, it is common to choose a double shear layer problem to simulate the formation of KHI in a periodic two-dimensional computational setting with unit side length. This stratified shear layer instability problem is used to demonstrate the evolution of linear per-

turbations into a transition to nonlinear two-dimensional hydrodynamic turbulence. The instability triggers small-scale vortical structures at the sharp density interface initially which eventually transitions through nonlinear interactions to a completely turbulent field.

## 2.2 Two-dimensional simulations

A two-dimensional implementation of the dual-shear layer KHI problem is devised through our aforementioned unstable perturbed compressible shear layer. This may be implemented through our computational domain which is a square of unit side length with the following initial conditions

$$\rho(x,y) = \begin{cases} 1.0, & \text{if } |y| \geq 0.25 \\ 2.0, & \text{if } |y| < 0.25 \end{cases} \tag{5}$$

$$u(x,y) = \begin{cases} \alpha, & \text{if } |y| \geq 0.25 \\ -\alpha, & \text{if } |y| < 0.25 \end{cases} \tag{6}$$

$$v(x,y) = \lambda \sin(2\pi n x/L) \tag{7}$$

$$p(x,y) = 2.5. \tag{8}$$

We can observe that the vertical component of the velocity is perturbed using a single-mode sine wave ($n = 2, L = 1$) with an amplitude $\lambda = 0.01$. Our two-dimensional numerical experiments are solved to a final dimensionless time of $t = 5$. We clarify that the $\mathbb{R}^2$ simulation domain for all experiments is set in $(x,y) \in [-0.5, 0.5] \times [-0.5, 0.5]$ with $N^2 = 16,384^2$ degrees of freedom. Fig. (1) represents a schematic expressing the initial conditions of our two-dimensional simulation. We remark, that in this study we perform implicit large eddy simulation (ILES) simulations by using a finite volume framework. Our numerical scheme utilizes the fifth-order accurate, weighted essential non-oscillatory (WENO) reconstructions equipped with the Roe's approximate Riemann solver (Roe, 1981) at the cell interfaces. It is well-known that the utilization of the artificial dissipation mechanism of ILES schemes (from the numerical viscosity of upwind biased state reconstructions) mimics the physical viscosity of the Navier–Stokes equations in the limit of infinite Reynolds numbers. We utilize a parallel approach for the computational solution of our governing laws implemented in the OpenMPI framework. Details about implementation and the computational performance of our solver may be found in Maulik and San (2017) additionally showing weak and strong scaling tests. Our three-dimensional simulations employ a similar approach.

Fig. (2) describes snapshots in time of the density field for this two-dimensional compressible turbulence test case when $\alpha = 1.0$. One can notice a transition to turbulence once an initial instability has developed. The shearing velocity magnitude given by $\alpha$ controls the compressibility which is apparent from comparisons with Fig. 3 and Fig. 4 where smaller values lead to formation of much smoother structures and consequently lead to shock-free fields in the incompressible limit. Evidence from Fig. (4) also shows a delay in the onset of turbulence due to a reduced shearing velocity. Table 1 also demonstrates the mean and maximum Mach number values at the final computational time $t = 5$. It is clear that the case for $\alpha = 0.25$ corresponds to perfectly subsonic regime with lower compressibility (i.e., the mean Mach number of $M = 0.15$).

Fig. (5) demonstrates the time evolution characteristics of the 2D KHI problem. On the left, we illustrate the time series of the domain integrated velocity amplitude (i.e., the root mean square values of the kinetic velocity) normalized with its initial condition with each $\alpha$ value. It is clear that the KHI instability starts earlier for larger $\alpha$ values. We also demonstrate the

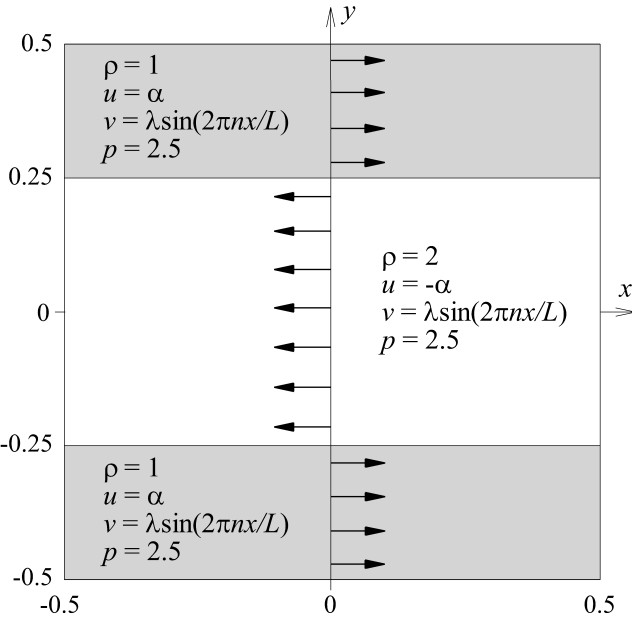

**Figure 1.** The stratified Kelvin-Helmholtz instability problem in a periodic square box of side length $L = 1$. Our initial condition reads as a single-mode perturbation to the $y$-component of the velocity to trigger the instability with $n = 2$ and the amplitude $\lambda = 0.01$. We extend this 2D domain along the $z$-direction to perform our 3D simulations in a triply-periodic domain with size $L$ in each side where we also use an initial perturbation to the $z$-component of the velocity given by $w = \lambda \sin(2\pi n z/L)$.

**Table 1.** The mean and maximum Mach numbers computed at final time $t = 5$.

| Resolution | $\alpha$ | $M_{mean}^{t=5}$ | $M_{max}^{t=5}$ |
|---|---|---|---|
| $16,384^2$ | 1.0 | 0.55 | 1.40 |
| $16,384^2$ | 0.5 | 0.30 | 1.28 |
| $16,384^2$ | 0.25 | 0.15 | 0.73 |

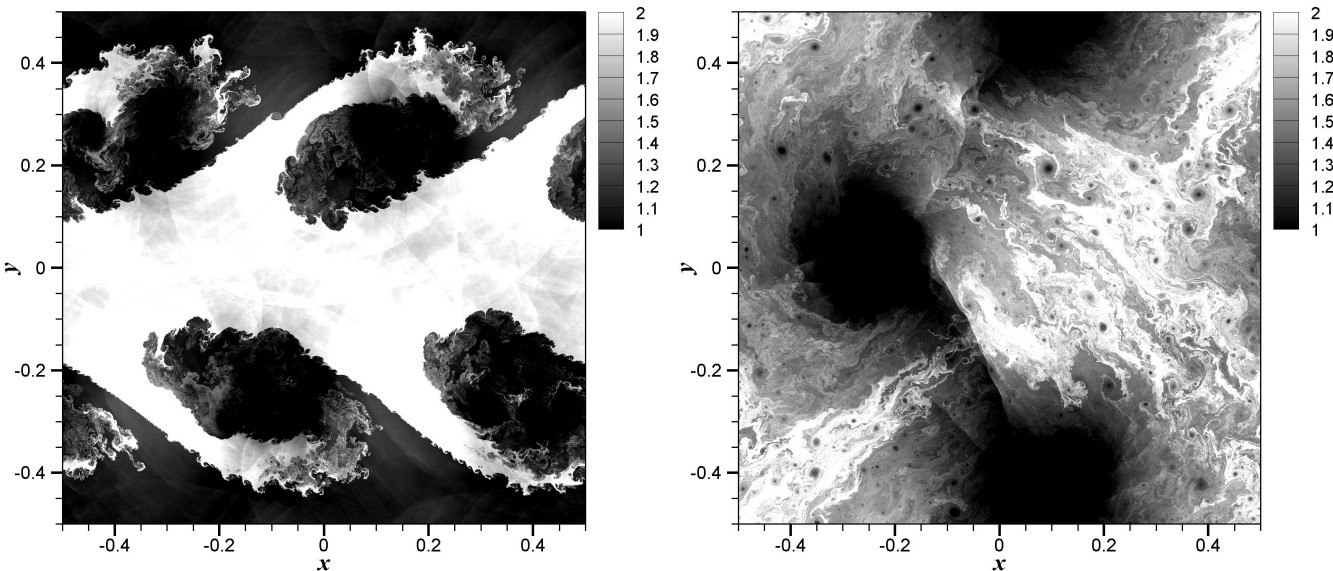

**Figure 2.** Time evolution of the density field for 2D KHI turbulence with $\alpha = 1.0$ demonstrating results at $t = 1$ (left) and $t = 5$ (right) obtained by a grid resolution of $N^2 = 16,384^2$.

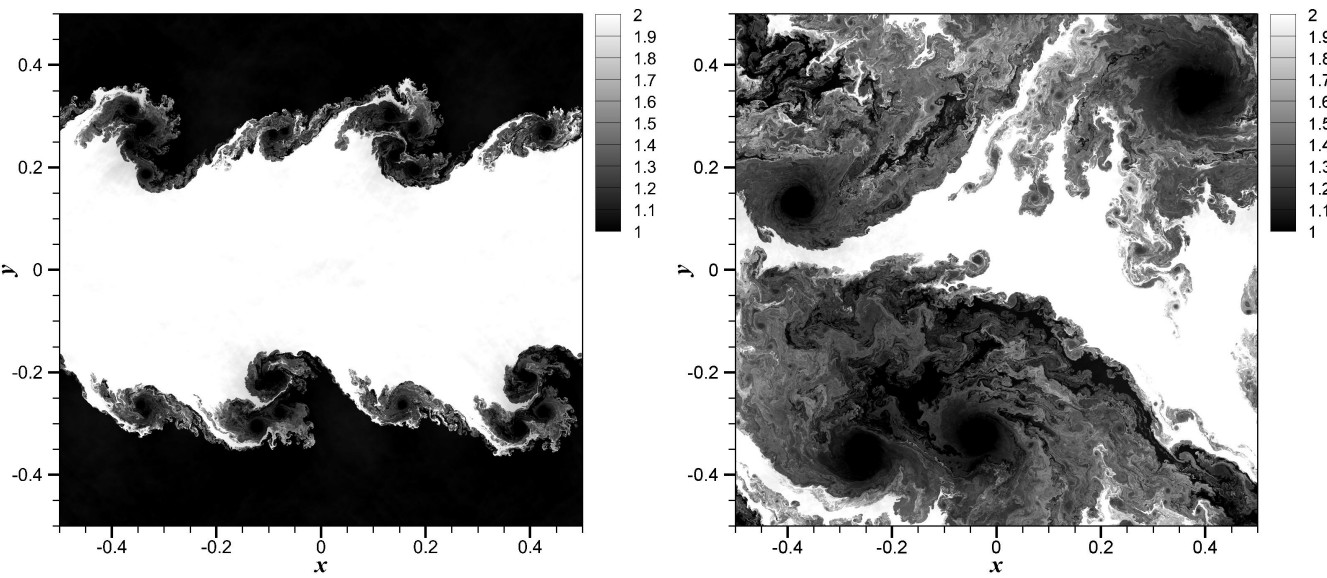

**Figure 3.** Time evolution of the density field for 2D KHI turbulence with $\alpha = 0.5$ demonstrating results at $t = 1$ (left) and $t = 5$ (right) obtained by a grid resolution of $N^2 = 16,384^2$.

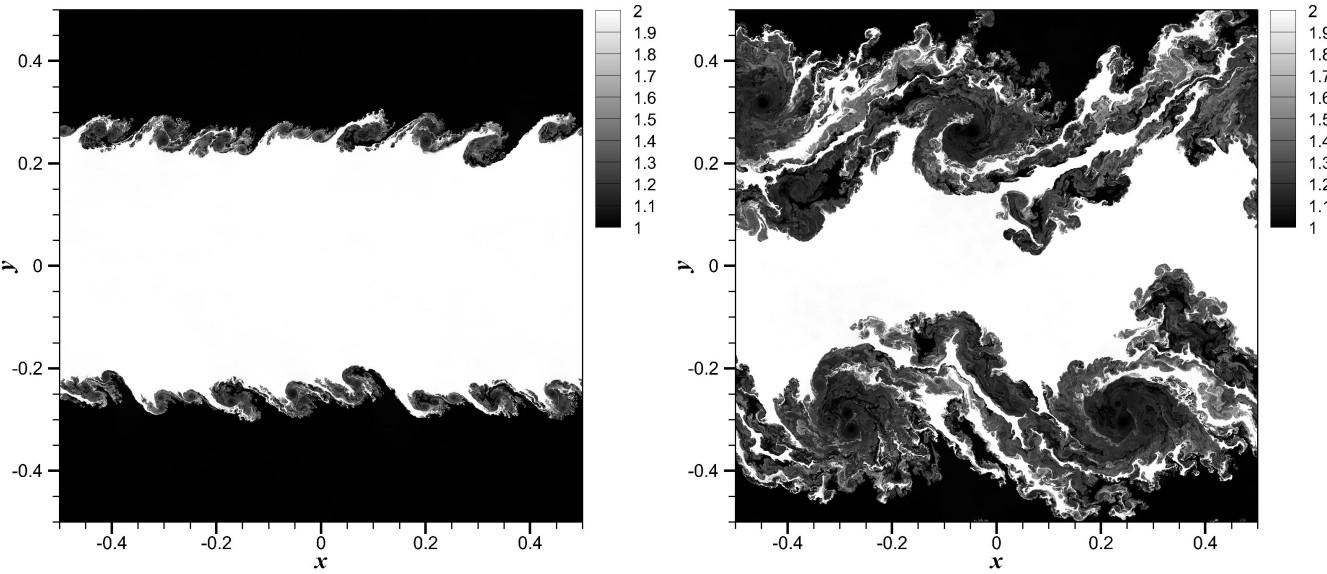

**Figure 4.** Time evolution of the density field for 2D KHI turbulence with $\alpha = 0.25$ demonstrating results at $t = 1$ (left) and $t = 5$ (right) obtained by a grid resolution of $N^2 = 16,384^2$.

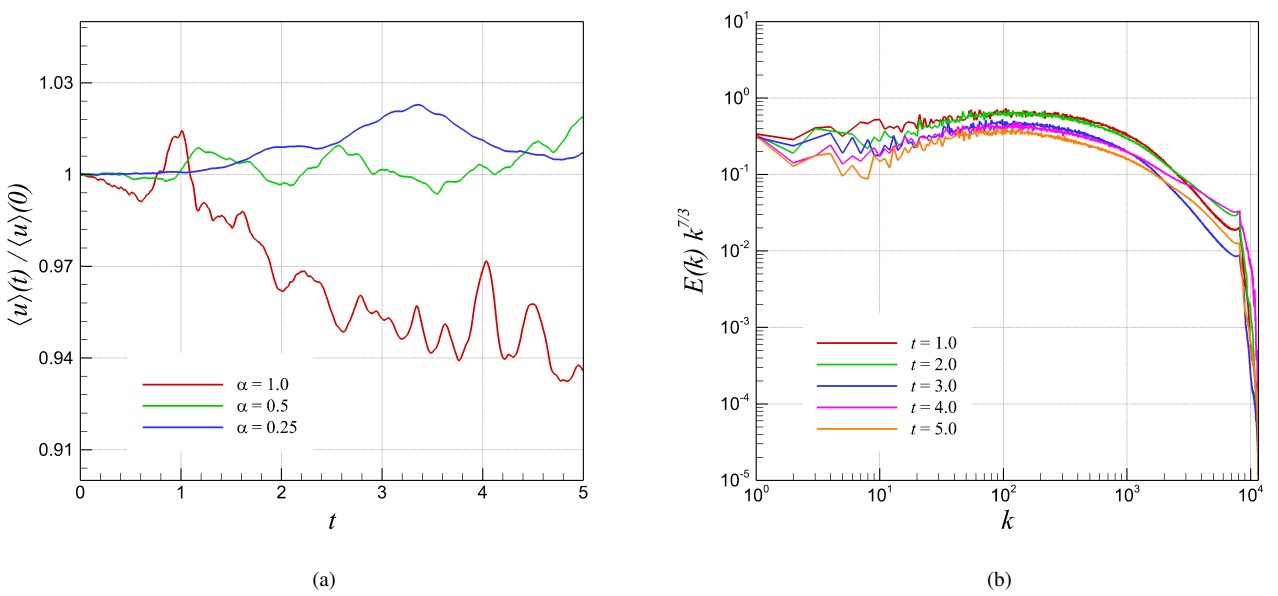

(a)

(b)

**Figure 5.** Time evolution of 2D KHI turbulence field characteristics with a resolution of $N^2 = 16,384^2$. (a) The normalized root mean square values of velocity $\boldsymbol{u}$ for various $\alpha$ values, and (b) compensated energy spectra computed from $\boldsymbol{u}$ at various times for $\alpha = 1.0$.

evolution of the compensated kinetic energy spectrum on the right for $\alpha = 1.0$. Similar statistical trends are observed at each time. Therefore, we will only focus on the results at the final time $t = 5$ in our statistical analysis presented in next section.

## 2.3  Three-dimensional simulations

While 2D compressible turbulence investigations are valuable for insight into the physical processes of systems which exhibit extreme aspect ratios (Boffetta and Ecke, 2012), it is well known that the process of energy transfer between scales is fundamentally different when compared to that of three-dimensional flows (Clercx and van Heijst, 2017). Isotropic, homogeneous, incompressible 3D turbulence is characterized by the famous Kolmogorov-Richardson cascade of energy where the largest vortices continuously inject energy into an inertial cascade which terminates in the Kolmogorov length scale (Kolmogorov, 1941) where viscous effects dissipate this energy. This is particularly applicable for engineering flows where it has been established that turbulence 'decays' in the absence of forcing due to viscous dissipation. In contrast, 2D turbulence exhibits the presence of an inverse energy cascade (given by Kraichnan-Bachelor-Leith theories (Kraichnan, 1967; Leith, 1971; Batchelor, 1969)) where energy from the smallest scales is transferred to largest scales. This has implications for the restoration of local isotropy (since large scale structures created by the inverse energy cascade affect the amount of enstrophy in the field and thus affect the energy dissipation rate). In the presence of a-periodic boundary conditions (a subject of future investigations), these newly created large scale structures may lead to significant alteration in scaling laws.

Our computational domain for the three-dimensional turbulence case is analogous to that of the two-dimensional domain. We utilize a domain given by a $\mathbb{R}^3$ set in $(x, y, z) \in [-0.5, 0.5] \times [-0.5, 0.5] \times [-0.5, 0.5]$ with $N^3 = 512^3$ degrees of freedom. Our initial conditions are given by

$$
\rho(x, y, z) = \begin{cases} 1.0, & \text{if } |y| \geq 0.25 \\ 2.0, & \text{if } |y| < 0.25 \end{cases} \tag{9}
$$

$$
u(x, y, z) = \begin{cases} \alpha, & \text{if } |y| \geq 0.25 \\ -\alpha, & \text{if } |y| < 0.25 \end{cases} \tag{10}
$$

$$
v(x, y, z) = \lambda \sin(2\pi n x / L) \tag{11}
$$

$$
w(x, y, z) = \lambda \sin(2\pi n z / L) \tag{12}
$$

$$
p(x, y, z) = 2.5. \tag{13}
$$

and periodic boundary conditions in all directions. We keep our parameters $n$, $L$ and $\lambda$ similar to those used in the two-dimensional case and utilize $N^3 = 512^3$ degrees of freedom for the simulation of our computational domain.

Fig. (6) shows the density field at times $t = 1$ and $t = 5$ for a shearing velocity magnitude of $\alpha = 1.0$. One can observe how the solution domain has transitioned almost entirely to a turbulent field for this case as against the very visible stratification observed in lower compressibility simulations given by $\alpha = 0.5$ and $\alpha = 0.25$ shown in Figs. (7) and (8), respectively. Our aim is to quantify the effect of the shearing velocity on the compressibility and scaling laws of these co-designed two- and

three-dimensional configurations. Similar to the 2D case, we have plotted the time evolution of the domain integrated velocity in Fig. (9) between $t = 0$ and $t = 5$. The decay rates in 3D simulations are substantially higher than those obtained in 2D simulations. This can be attributed to the use of a lesser number of grid point sampled in each direction. However, the energy spectra trend is similar and yields a $k^{-5/3}$ spectrum at each time. In the following section, we thus present a systematic analysis based on data obtained at $t = 5$.

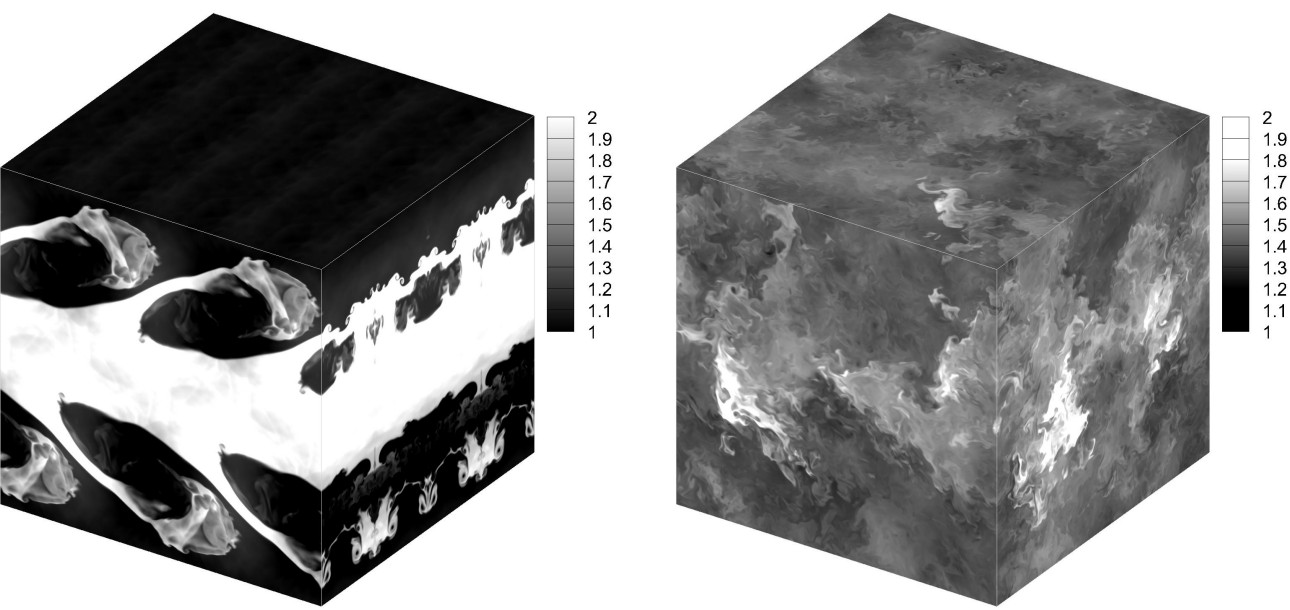

**Figure 6.** Time evolution of the density field for 3D KHI turbulence with $\alpha = 1.0$ demonstrating results at $t = 1$ (left) and $t = 5$ (right) obtained by a grid resolution of $N^3 = 512^3$.

## 3 Turbulence statistics and scaling exponents

### 3.1 Kinetic energy spectrum

The first statistical measure we investigate is given by the classical kinetic energy spectra. To obtain this spectra, we start with an expression for the spatial kinetic energy in wavenumber space given by (Kida et al., 1990)

$$E(\mathbf{k}, t) = \frac{1}{2} |\hat{\mathbf{u}}(\mathbf{k}, t)|^2 \tag{14}$$

where $\hat{\mathbf{u}}(\mathbf{k}, t)$ is the Fourier transform of the velocity vector in the wavenumber space. Eq. (14) can be also rewritten in terms of velocity components (assuming a two-dimensional Cartesian domain) as

$$E(\mathbf{k}, t) = \frac{1}{2} \left( |\hat{u}(\mathbf{k}, t)|^2 + |\hat{v}(\mathbf{k}, t)|^2 \right) \tag{15}$$

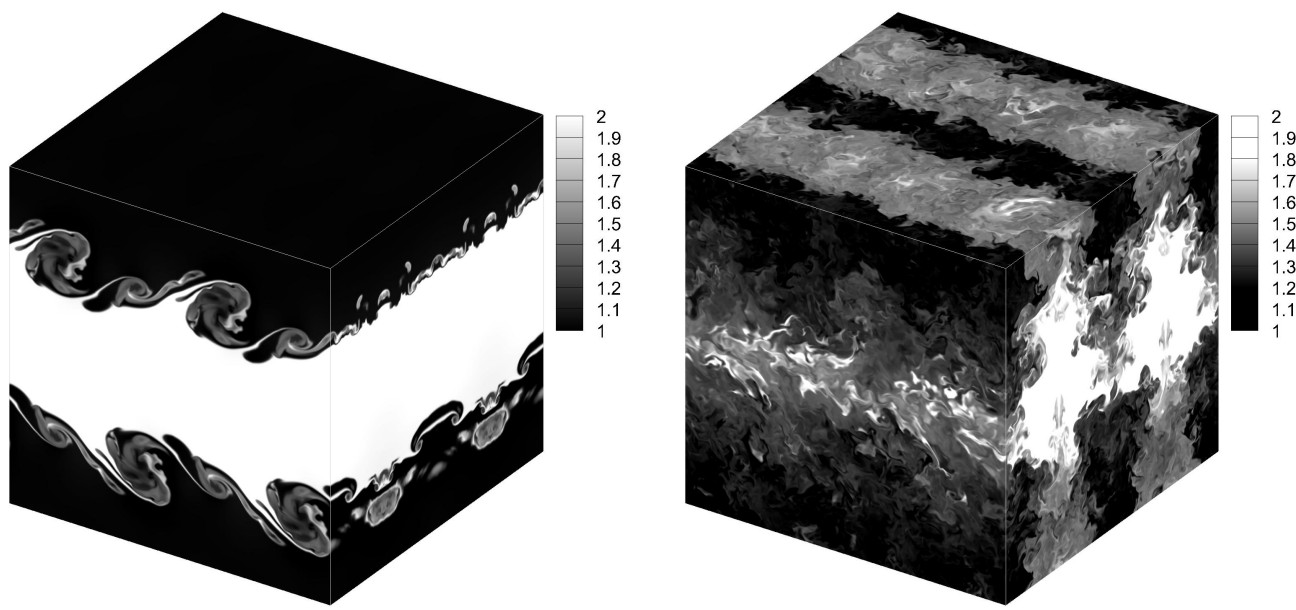

**Figure 7.** Time evolution of the density field for 3D KHI turbulence with $\alpha = 0.5$ demonstrating results at $t = 1$ (left) and $t = 5$ (right) obtained by a grid resolution of $N^3 = 512^3$.

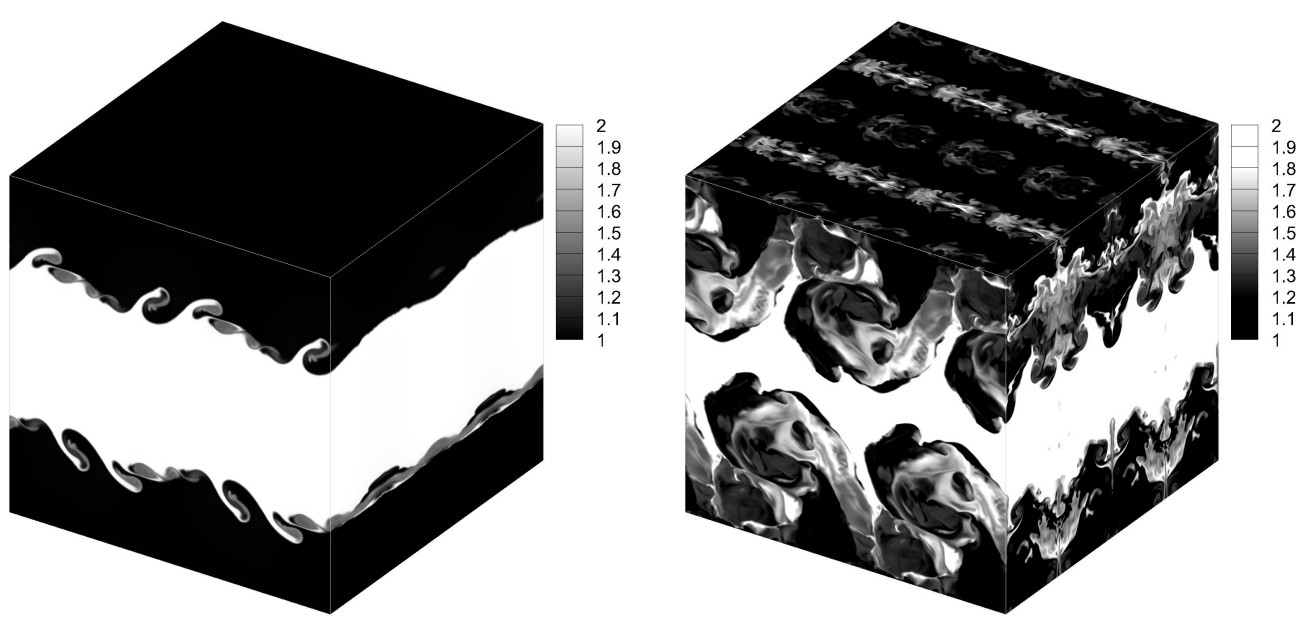

**Figure 8.** Time evolution of the density field for 3D KHI turbulence with $\alpha = 0.25$ demonstrating results at $t = 1$ (left) and $t = 5$ (right) obtained by a grid resolution of $N^3 = 512^3$.

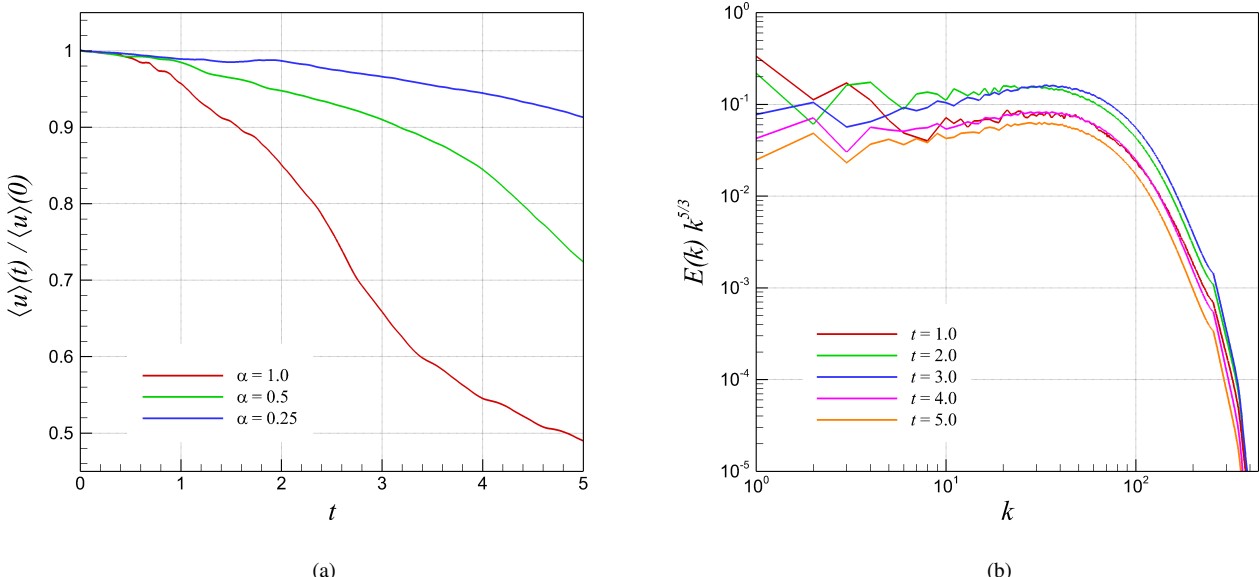

(a)

(b)

**Figure 9.** Time evolution of 3D KHI turbulence field characteristics with a resolution of $N^3 = 512^3$. (a) The normalized root mean square values of velocity $u$ for various $\alpha$ values, and (b) compensated energy spectra computed from $u$ at various times for $\alpha = 1.0$.

where we compute velocity components $\hat{u}(\mathbf{k}, t)$ and $\hat{v}(\mathbf{k}, t)$ using a fast Fourier transform algorithm (Press et al., 1996). Finally, the spectra can be calculated by integrating over a unit bandwidth (i.e., angle-averaged) in the following manner

$$E(k,t) = \sum_{k-\frac{1}{2} \leq |\acute{\mathbf{k}}| < k+\frac{1}{2}} \boldsymbol{E}(\acute{\mathbf{k}}, t) \tag{16}$$

where $k = |\mathbf{k}| = \sqrt{k_x^2 + k_y^2}$ in $\mathbb{R}^2$. Extensions to three-dimensions are straightforward.

5 ## 3.2 Density-weighted kinetic energy spectrum

The kinetic energy spectrum is generally utilized for characterizing the energy content of scales in incompressible turbulent flows and does not take the localized scale content of the density into consideration, the localized scale content of the density. To include these density effects, following Lele (1994); Kritsuk et al. (2007), we define an energy spectrum built on density-weighted velocity $\boldsymbol{\omega} = \sqrt{\rho}\boldsymbol{u}$, i.e., though using

10 $$\boldsymbol{E}(\mathbf{k}, t) = \frac{1}{2}|\hat{\boldsymbol{\omega}}(\mathbf{k}, t)|^2, \tag{17}$$

where we can apply the same angle-averaged rule given by Eq. (16) to obtain one-dimensional spectra.

Fig. (10) describes the spherical-averaged energy spectra for the three-dimensional test case. Note here that the spherical average implies that the local energy content in the Fourier domain is integrated over a spherical shell of radius $k$ in three

dimensions. One can observe a scaling behavior that corresponds to classical Kolmogorov theory in the infinite Reynolds number limit (i.e., an inertial range with $k^{-5/3}$ scaling) for both purely kinetic energy spectra and density-weighted kinetic energy spectra. The finer dissipative scales are seen to display a $k^{-6}$ scaling behavior for both these statistical quantities as well. We have also plotted the compensated energy spectra, which illustrate the scaling laws more quantitatively following the horizontal lines.

The data presented in Fig. (10) has been obtained by performing a 3D fast Fourier transform (FFT) procedure. As a practical implementation point of view, we perform a slightly different approach to compute energy spectra. The main advantage of this procedure is that it is naturally suited to any parallel computing architecture. For an analogy with the two-dimensional test cases, we present a transversely-averaged energy spectra in Fig. (11) wherein the circular averaging of the energy in the Fourier domain is carried out over different two-dimensional $z$ planes which are then spatially averaged over the depth of the domain. Similar trends to the spherical averaging spectral scaling are observed for this case. However, we note that the obtained spectra are less noisy when using a direct 3D FFT procedure. This can be interpreted by the quasi-homogeneity of the flow after the onset of turbulence.

We investigate the performance of the same metrics for the two-dimensional test case and obtain scaling behavior as seen in Fig. (12) where a $k^{-3}$ scaling behavior is obtained in accordance with the direct-cascade of enstrophy espoused by Kraichnan-Batchelor-Leith (KBL) theory for the inertial range especially for the lower compressiblity ratio. A higher magnitude of $\alpha$ is seen to yield a more flatten spectrum towards $k^{-7/3}$ scaling and also delay the formation of the $k^{-6}$ cascade in the dissipation range. Fig. (12) also shows the spectral scaling obtained from the density-weighted kinetic energy spectra where scaling behavior corresponding to $k^{-7/3}$ is seen for all $\alpha$ values. This suggests that the two-dimensional configuration of the test-case is affected by the packaging of density content at different scales. The dissipation zone shows a similar behavior using this metric where a delay in scaling with $k^{-6}$ is obtained by an increase in the magnitude of $\alpha$. We can conclude that the density-weighted spectrum becomes a more universal representation for various degrees of compressibility.

Fig. (13) shows the effect of the parameter $\alpha$ on the compressibility of the two-dimensional turbulence case through the use of a compensated energy spectra where the distance from the origin in the Fourier space (in other words $k$) is used to weight instantaneous energy content. We only present the compensated energy distribution in the first quadrant of the Fourier space. At $\alpha = 1.0$ one can observe a distinct loss of isotropy in the energy content of the solution field (in spectral space) which corresponds to an enhanced compressibility. In comparison, $\alpha = 0.5$ and $\alpha = 0.25$ display a behavior which is rather isotropic in nature indicating weak compressibility.

To demonstrate the effect of density more clearly, we present the difference spectra for the 2D KHI turbulence in Fig. (14). Here, we compute the spectrum of the difference between the velocity $\boldsymbol{u}$ and the normalized density-weighted velocity $\sqrt{\rho}\boldsymbol{u}/\langle\sqrt{\rho}\rangle$, where $\sqrt{\rho}$ refers to the spatial average of square root of density. The results show a clear inertial range with the $k^{-5/3}$ scaling. This is a manifestation of the density effect in 2D KHI turbulence.

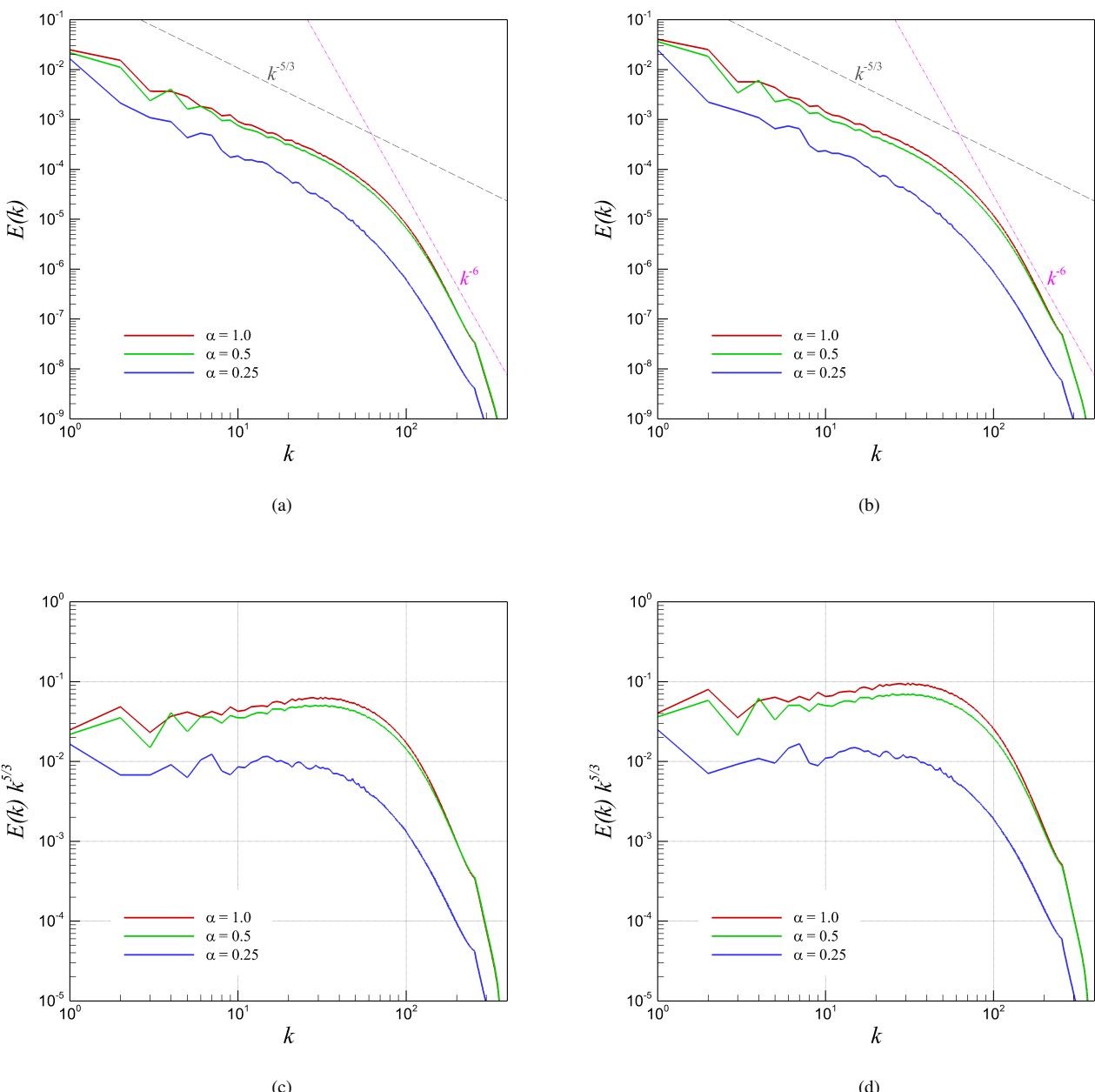

**Figure 10.** Spherical-averaged energy spectra for 3D KHI turbulence. (a) spectra built on using the velocity $\boldsymbol{u}$, (b) spectra built on using the density-weighted velocity $\boldsymbol{\omega} = \sqrt{\rho}\boldsymbol{u}$, (c) compensated spectra built on using the velocity $\boldsymbol{u}$, and (d) compensated spectra built on using the density-weighted velocity $\boldsymbol{\omega} = \sqrt{\rho}\boldsymbol{u}$.

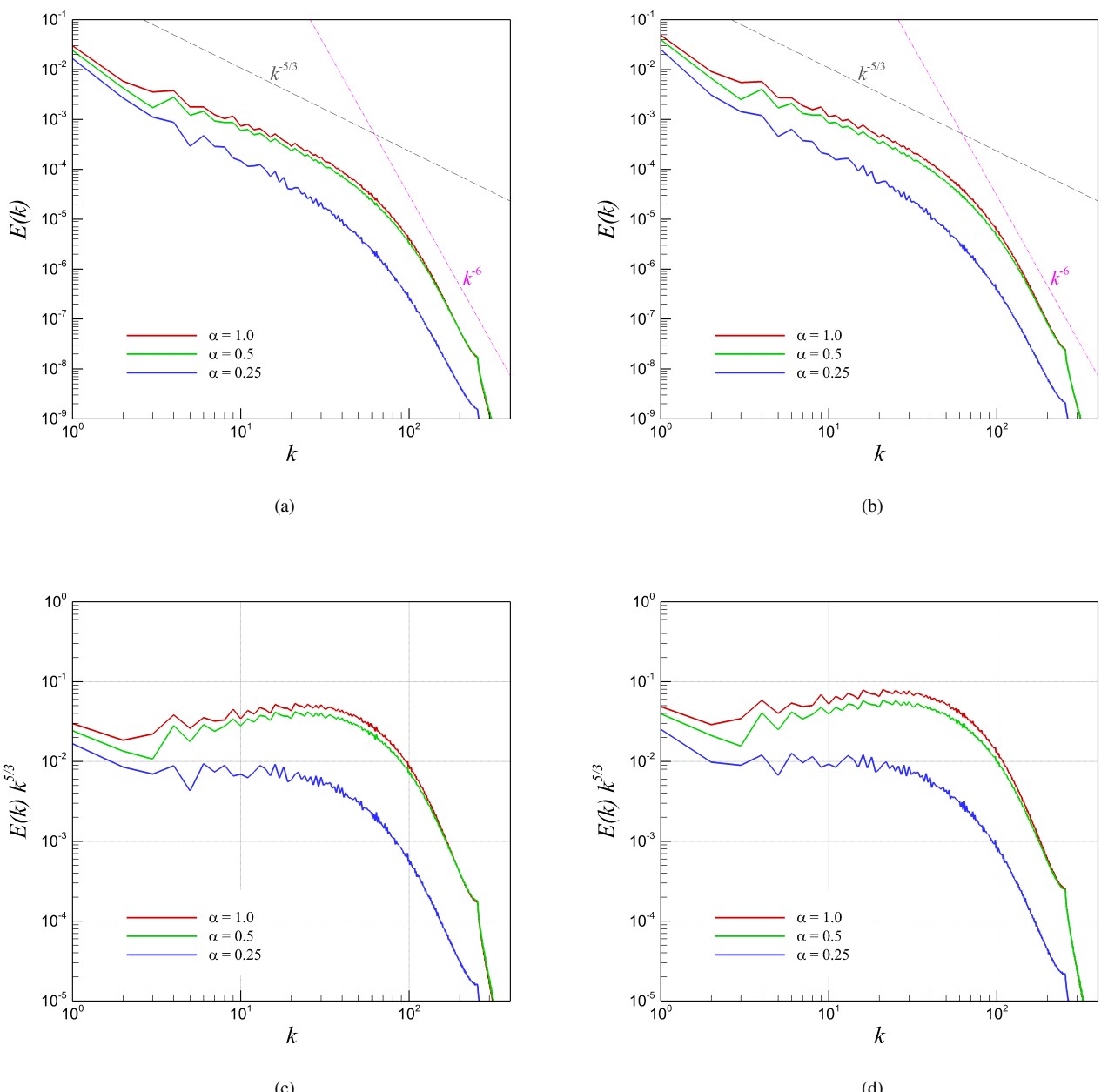

**Figure 11.** Transversely-averaged energy spectra for 3D KHI turbulence. An angle-averaged kinetic energy spectrum is first computed at each $z$-plane using a two-dimensional FFT transform and then followed by a spatial averaging procedure along the $z$-direction. (a) spectra built on using the velocity $\boldsymbol{u}$, (b) spectra built on using the density-weighted velocity $\boldsymbol{\omega} = \sqrt{\rho}\boldsymbol{u}$, (c) compensated spectra built on using the velocity $\boldsymbol{u}$, and (d) compensated spectra built on using the density-weighted velocity $\boldsymbol{\omega} = \sqrt{\rho}\boldsymbol{u}$.

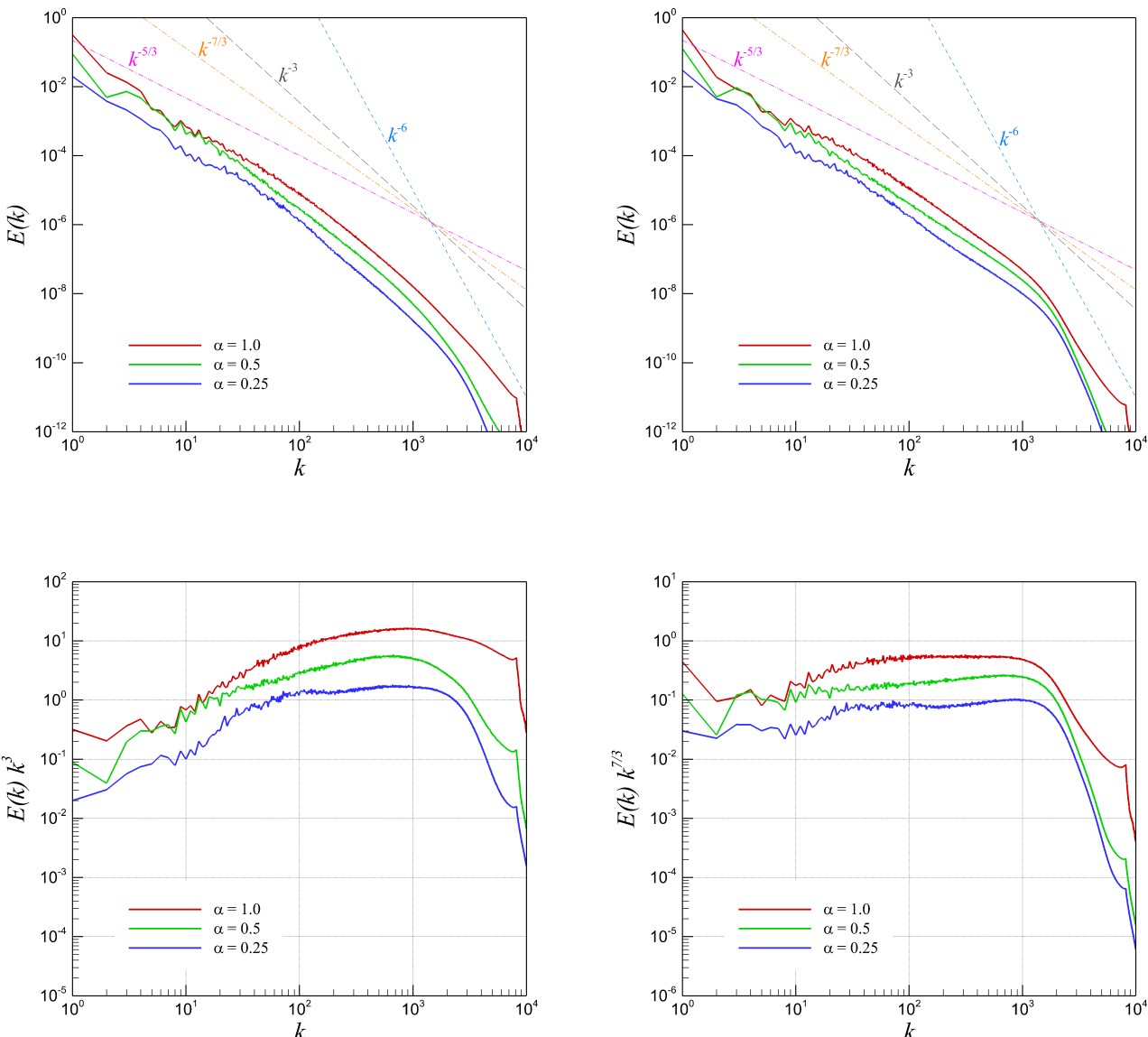

**Figure 12.** Angle-averaged energy spectra for 2D KHI turbulence. (a) Spectra built on using the velocity $\boldsymbol{u}$, (b) spectra built on using the density-weighted velocity $\boldsymbol{\omega} = \sqrt{\rho}\boldsymbol{u}$, (c) compensated spectra built on using the velocity $\boldsymbol{u}$, and (d) spectra built on using the density-weighted velocity $\boldsymbol{\omega} = \sqrt{\rho}\boldsymbol{u}$.

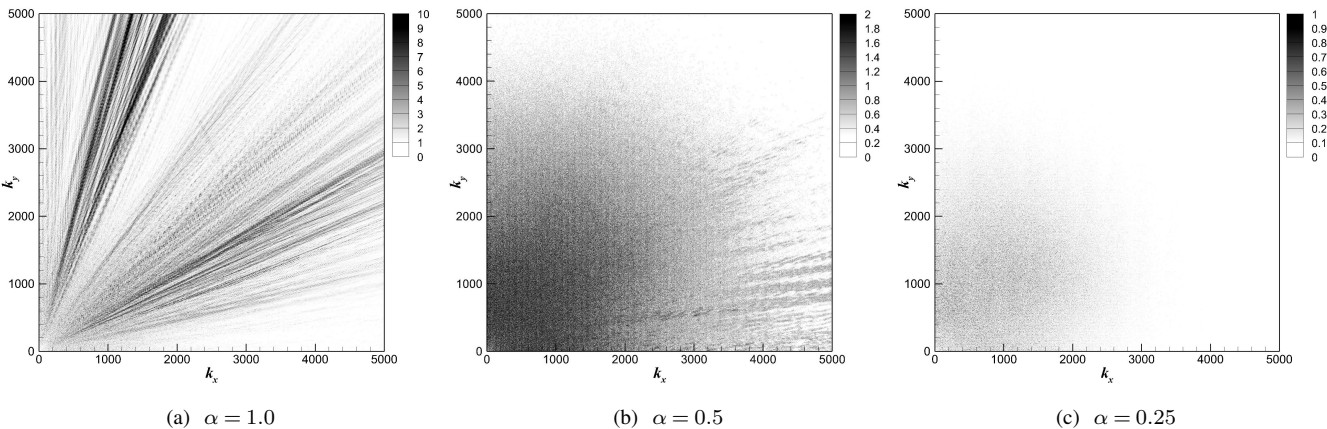

**Figure 13.** Compensated, $k^4 E(\mathbf{k}, t = 5)$, kinetic energy spectra for 2D KHI turbulence for various $\alpha$ values.

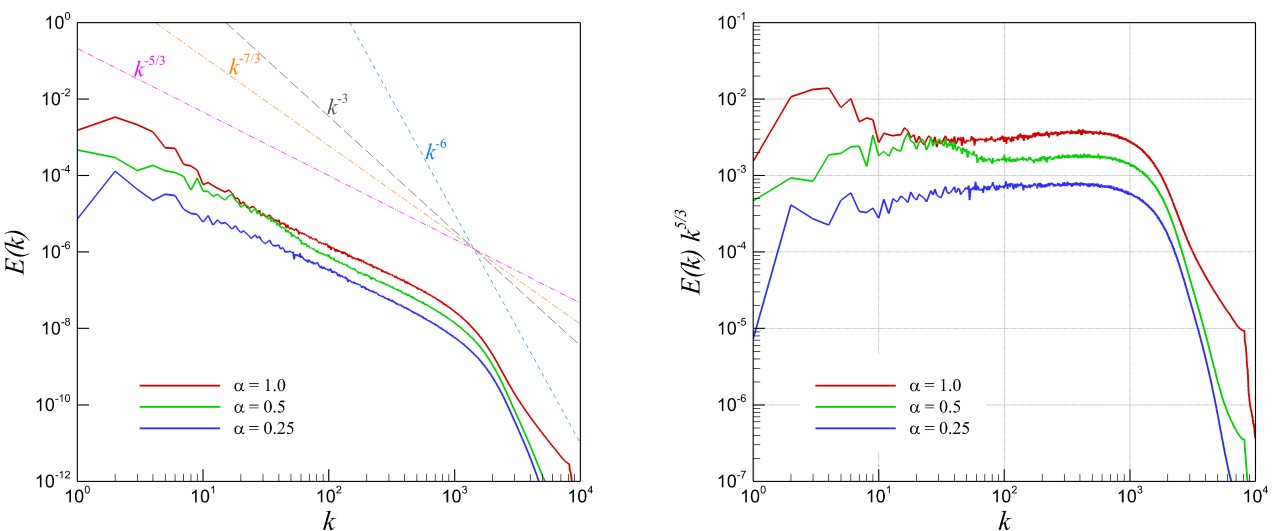

**Figure 14.** The difference spectra for 2D KHI turbulence. (a) Difference spectra between the kinetic velocity field and the normalized density-weighted velocity field (i.e., $E(k)$ is obtained from $\boldsymbol{g} = \boldsymbol{u} - \sqrt{\rho}\boldsymbol{u}/\langle\sqrt{\rho}\rangle$ vector field), and (b) its compensated representation.

### 3.3 Helmholtz decomposition

To study the effect of compressibility in more detail we perform the Helmholtz decomposition to compute energy spectra from the curl-free and divergence-free components of the velocity field. This decomposition has been extensively used in turbulence studies (i.e., see Sagaut and Cambon (2008); Jagannathan and Donzis (2016); Wang et al. (2017); Falkovich and Kritsuk (2017); Wang et al. (2018)). In our present work, we investigate the behavior of energy spectra using both the kinematic velocity and density-weighted velocity fields in 2D and 3D KHI turbulence problems. Let $v$ be a vector field in $\in \mathbb{R}^n$ (e.g., $v$ could be the kinetic velocity field $u$ or the density-weighted velocity field $\omega = \sqrt{\rho}u$), then $v$ can be decomposed into a curl-free and a divergence-free component (Aris, 2012):

$$v = \nabla\phi + \nabla \times A, \tag{18}$$

which can be rewritten as

$$v = v^C + v^S, \tag{19}$$

where $v^c = \nabla\phi$ is the compressive (curl-free) component since the curl of a gradient of any scalar field $\phi$ is zero, and $v^s = \nabla \times A$ is the solenoidal (divergence-free) component since the divergence of a curl of any vector field $A$ is zero. Taking the divergence of Eq. (18) yields the following Poisson equation

$$\nabla . v = \nabla^2 \phi, \tag{20}$$

which can be solved for $\phi$ efficiently using an FFT procedure since $v$ is provided as a quantity of interest that we would like to decompose into two parts. Once $\phi$ is computed, the compressive and solenoidal parts can be easily computed as follows:

$$v^c = \nabla\phi, \tag{21}$$

$$v^s = v - v^c. \tag{22}$$

We note that there would be infinitely many candidates for the compressive component since the multiplication of $\phi$ with any arbitrary constant after solving the Poisson equation would still yield a curl-free velocity field. However, the energy spectrum scaling behaviors would remain identical for each realization.

Fig. (15) presents the compensated energy spectra for the 3D KHI problem using both definitions of the velocity vector field (i.e., the kinematic velocity and the density-weighted velocity). We have obtained a $k^{-5/3}$ dominant scaling for the solenoidal component in both definitions. However, the compressive component demonstrates an anomalous spectrum especially when we use the kinetic velocity definition. This anomaly can be also linked to the results of the pressure power spectra that we present in next section. Fig. (16) presents the same analysis for the case of 2D KHI turbulence. Both compressive and solenoidal components scale with the $k^{-5/3}$ slope for the density-weighted velocity field. However, there is a clear difference for the results with various values of $\alpha$ when we look at the Helmholtz decomposition of the kinetic velocity field. The solenoidal

inertial range scaling becomes $k^{-3}$ for lower $\alpha$ values, which is consistent with Kraichnan theory. However, the scaling steepens and gets closer towards $k^{-2}$ for increasing $\alpha$, which is also consistent with the Kadomtsev–Petviashvili spectrum for acoustic turbulence.

## 3.4 Density power spectrum

Observations on the density power spectrum have played an important role in astrophysics applications (Armstrong et al., 1981). Although it has been established that the density power spectrum has an inertial scaling of $k^{-5/3}$ (Shaikh and Zank, 2010; Donzis and Jagannathan, 2013), similar to the Kolmogorov energy spectrum, Bayly et al. (1992) demonstrated that it depends on the flow regime as well as the initial conditions by considering a three-dimensional weakly compressible hydrodynamic turbulence setup. By studying weakly compressible two-dimensional flows, Terakado and Hattori (2014) showed that

the density spectrum scales between $k^{-1}$ and $k^{-5}$ for nonuniform and uniform entropy cases, respectively. They presented a great discussion for state-of-the-art computations and scaling law observations for the density power spectrum.

In order to quantify the effect of the scale content of density alone, we devise a power spectrum that reflects the average packaging of density over different scales at any given time in the simulation. This may be given by the following expression

$$\mathbf{\Gamma}(\mathbf{k},t) = \frac{1}{2}|\hat{\rho}(\mathbf{k},t)|^2, \tag{23}$$

followed by angle averaging which leads to

$$\Gamma(k,t) = \sum_{k-\frac{1}{2} \leq |\acute{\mathbf{k}}| < k+\frac{1}{2}} \mathbf{\Gamma}(\acute{\mathbf{k}},t). \tag{24}$$

Observations regarding the difference in scaling behavior of the kinetic energy and density-weighted kinetic energy spectra give us a cause to compare the scaling behavior of the density power spectra for both our two- and three-dimensional test cases. Fig. (17) shows the density power spectra for the three-dimensional turbulence test case where it can be seen that a

five-thirds law is followed for the arrangement of density content in the solution field. A dissipation range scaling of $k^{-6}$ can also be observed. It can be seen that the variation of parameter $\alpha$ does not seem to affect scaling behavior appreciably. Fig. (18) shows a similar examination for the two-dimensional test case where a considerable difference in scaling behavior is observed. The imposition of two-dimensional turbulence leads to a considerable alteration in the scaling behavior of the density power spectrum with a $k^{-5/3}$ scaling observed in the inertial range and a $k^{-3}$ scaling in the dissipation range. In fact, this packaging

of density consequently affects the density-weighted kinetic energy spectra described in Fig. (12). The intercomparison of the two- and three-dimensional statistical quantities suggests that the density power spectrum (i.e., the arrangement of density at different wavenumbers) plays an important role with increased compressibility of any simulation wherein the $k^{-5/3}$ scaling causes a deviation from $k^{-3}$ scaling associated with two-dimensional incompressibility to $k^{-7/3}$ scaling for $\alpha = 1.0$ for the same test-case. In contrast, the $k^{-5/3}$ density power spectrum of three-dimensional turbulence causes no variation in scaling

behavior with increased compressibility and also causes similar scaling behaviors for both averaged kinetic energy spectra as well as averaged density-weighted kinetic energy spectra as seen in Fig. (10). This is one of the central conclusions of this investigation.

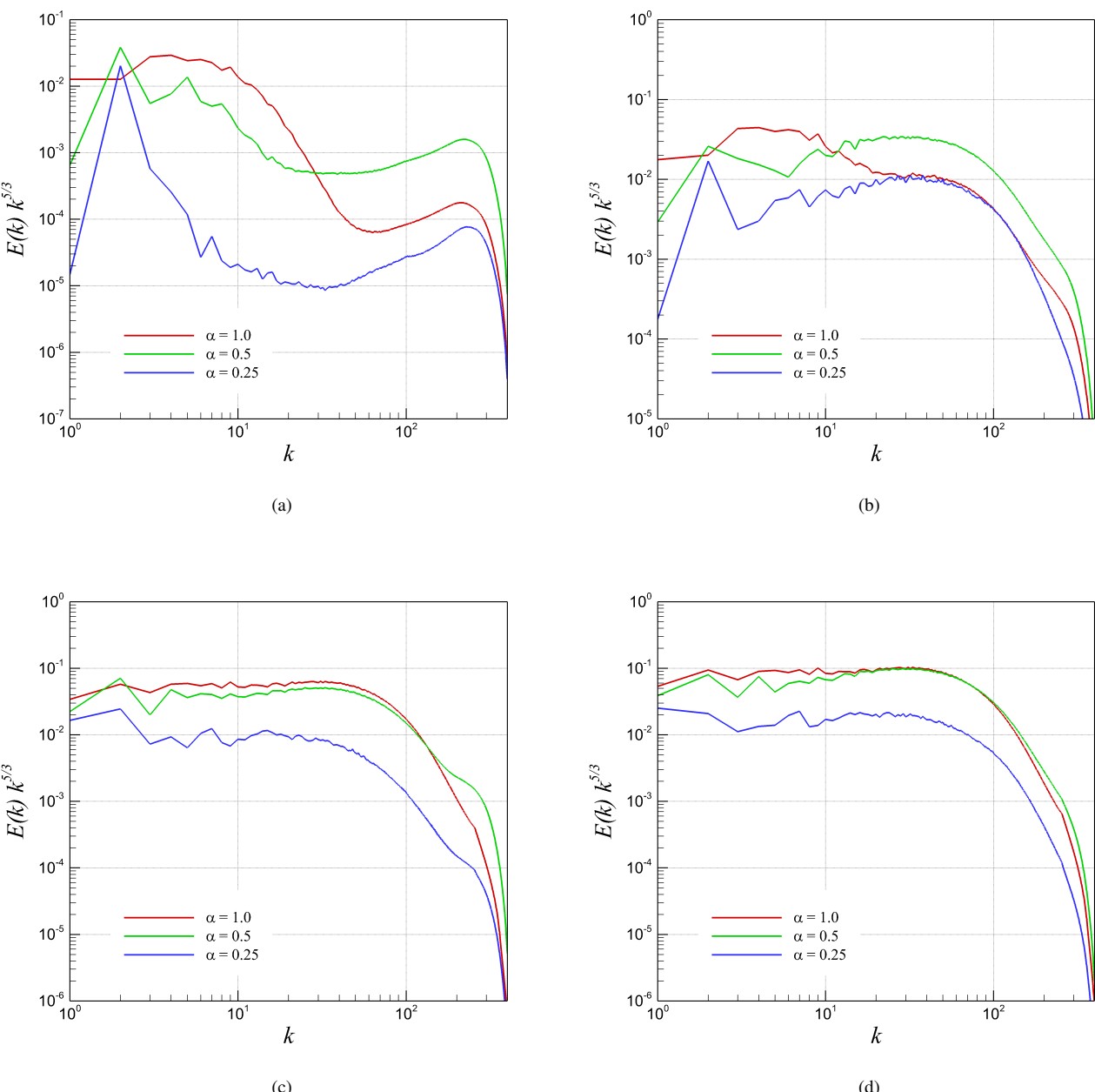

**Figure 15.** Helmholtz decomposition of energy spectra into compressive (curl-free) and solenoidal (divergence-free) parts for 3D KHI turbulence. (a) Compensated compressive spectra from $\boldsymbol{u}$, (b) compensated compressive spectra from $\boldsymbol{\omega} = \sqrt{\rho}\boldsymbol{u}$, (c) compensated solenoidal spectra from $\boldsymbol{u}$, and (d) compensated solenoidal spectra from $\boldsymbol{\omega} = \sqrt{\rho}\boldsymbol{u}$.

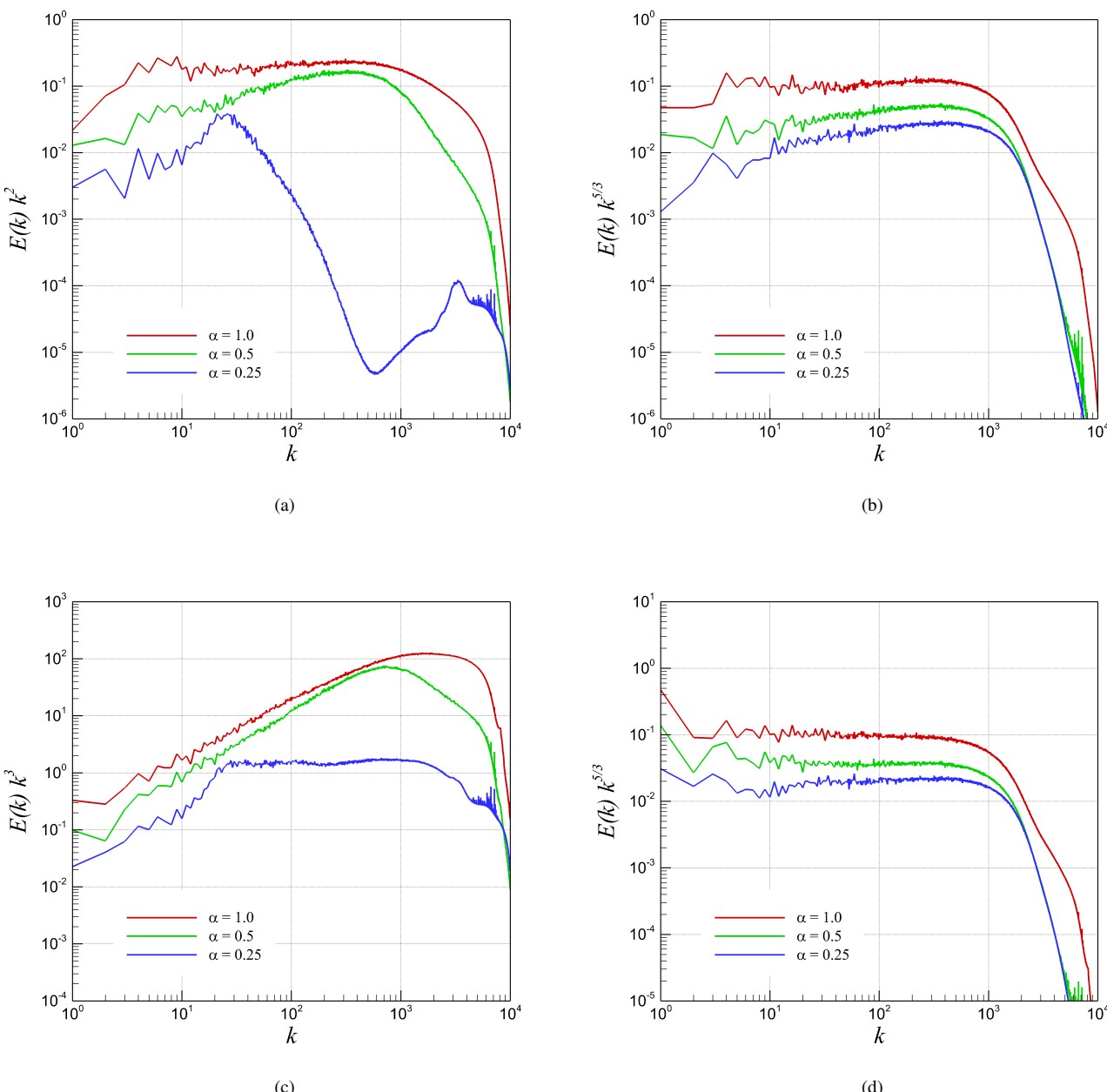

**Figure 16.** Helmholtz decomposition of energy spectra into compressive (curl-free) and solenoidal (divergence-free) parts for 2D KHI turbulence. (a) Compensated compressive spectra from $\boldsymbol{u}$, (b) compensated compressive spectra from $\boldsymbol{\omega} = \sqrt{\rho}\boldsymbol{u}$, (c) compensated solenoidal spectra from $\boldsymbol{u}$, and (d) compensated solenoidal spectra from $\boldsymbol{\omega} = \sqrt{\rho}\boldsymbol{u}$.

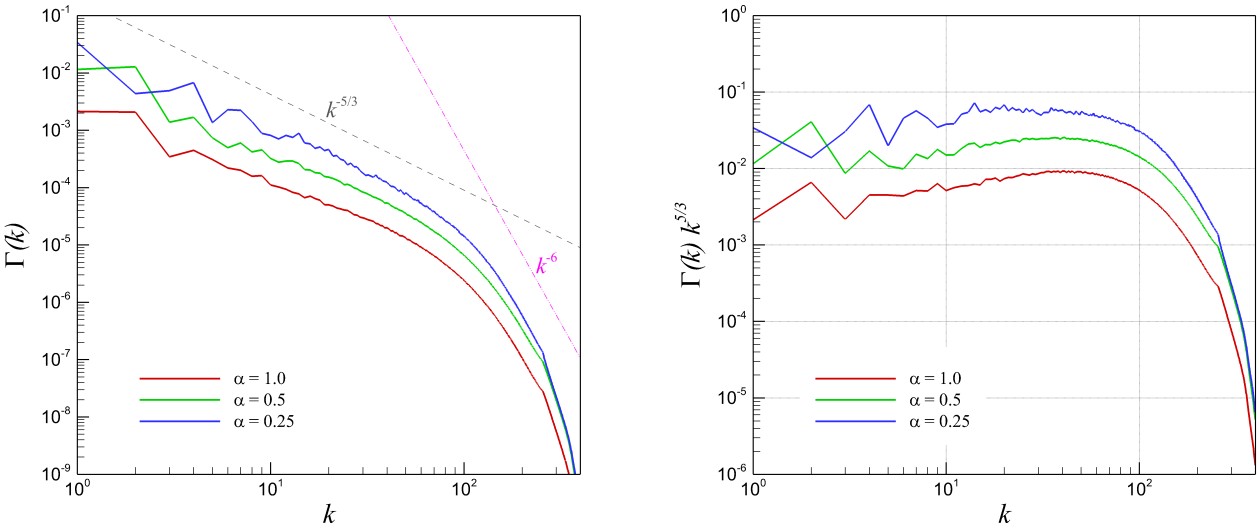

**Figure 17.** Spherical-averaged density power spectra for 3D KHI turbulence (left) and its compensated form (right).

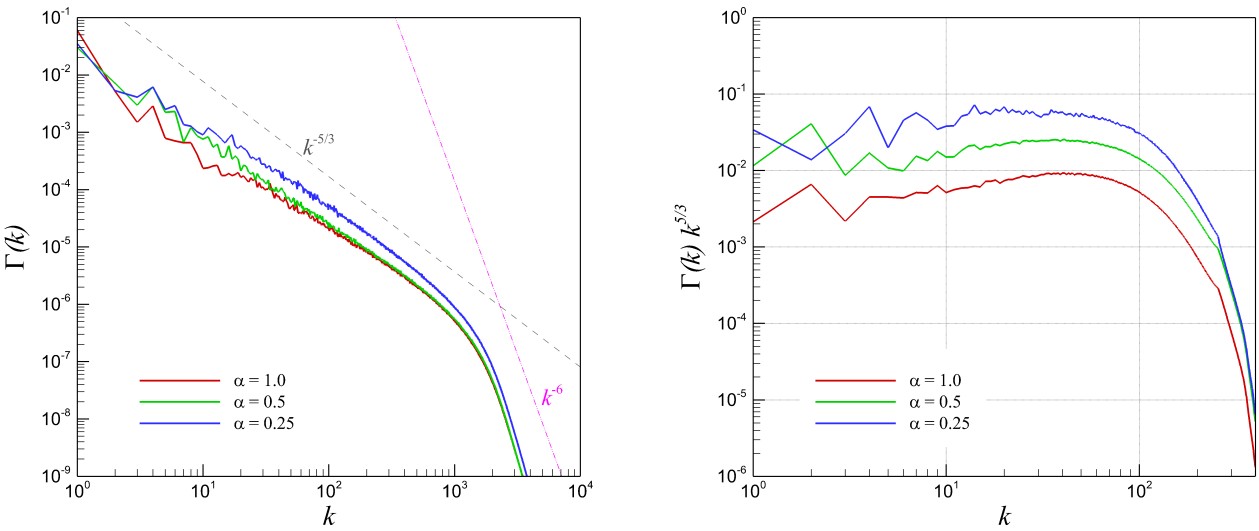

**Figure 18.** Angle-averaged density power spectra for 2D KHI turbulence (left) and its compensated form (right).

## 3.5 Pressure power spectrum

Similar to the density power spectrum defined in Eq. (25), the pressure power spectrum can be computed as:

$$\Pi(\mathbf{k},t) = \frac{1}{2}|\hat{p}(\mathbf{k},t)|^2, \tag{25}$$

and its angle-averaged form reads as

$$\Pi(k,t) = \sum_{k-\frac{1}{2}\leq|\acute{\mathbf{k}}|<k+\frac{1}{2}} \Pi(\acute{\mathbf{k}},t). \tag{26}$$

As discussed in Lesieur et al. (1999), the pressure spectrum can be expressed by $\Pi(k) \propto kE(k)^2$ by considering dimensional arguments. Indeed, this yields a pressure spectra scaling of $k^{-7/3}$ for the Kolmogorov regime and a pressure spectra scaling of $k^{-5}$ for the Kraichnan regime. Fig. (19) and Fig. (20) demonstrate the pressure power spectra for the 3D and 2D KHI problems, respectively. In the 3D case, it is clear that our results are consistent with the theoretical estimate of $k^{-7/3}$ scaling for all values of the compressibility parameter $\alpha$. However, in 2D turbulence we only observe $k^{-5}$ scaling for smaller scales (i.e., higher wave numbers). Particularly for weaker compressibility, given by the $\alpha = 0.25$ case, the $k^{-5}$ scaling starts earlier. Fig. (20) clearly illustrates that the pressure power spectrum inertial scaling becomes $k^{-5/3}$ for stronger compressibility. These results indicate that the pressure power spectrum can be a useful tool to characterize two-dimensional compressible turbulence.

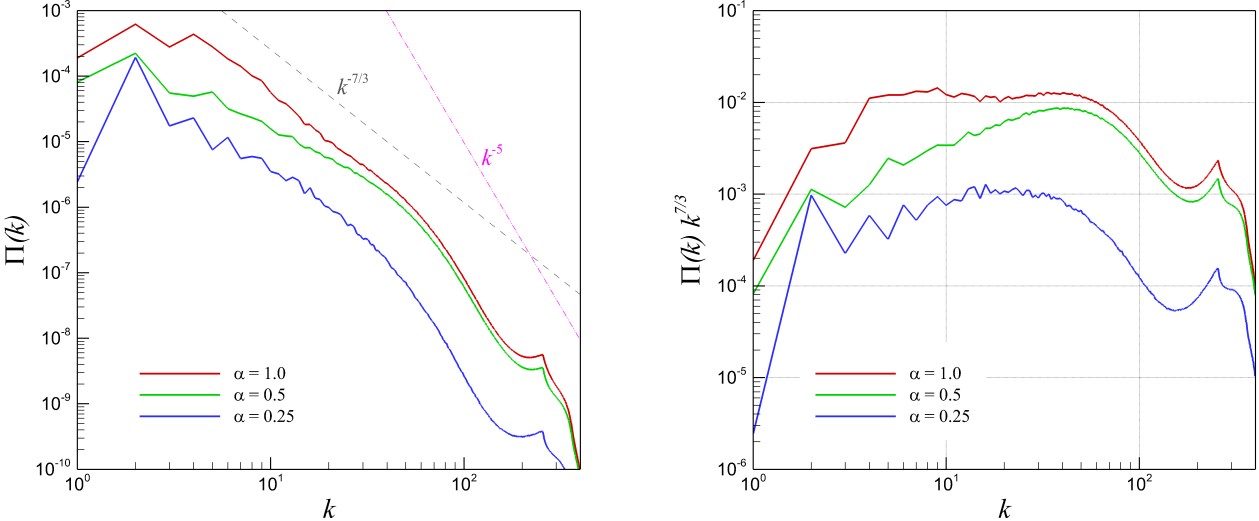

**Figure 19.** Spherical-averaged pressure power spectra for 3D KHI turbulence (left) and its compensated form (right).

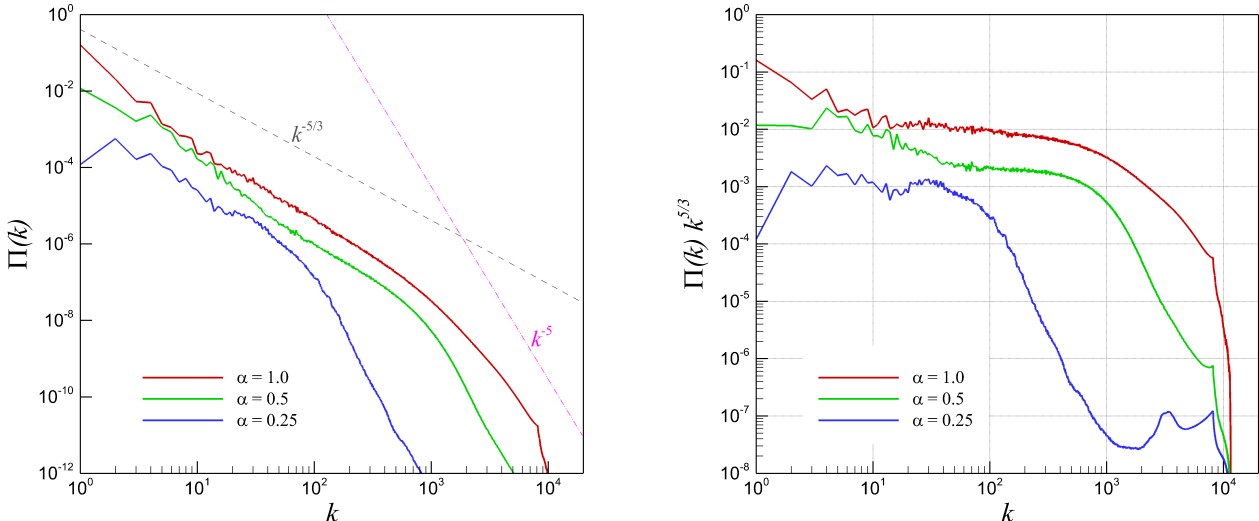

**Figure 20.** Angle-averaged pressure power spectra for 2D KHI turbulence (left) and its compensated form (right).

### 3.6 Velocity structure functions

Statistical inferences about the nature of compressible turbulence may also be drawn through the use of velocity structure functions which also show scaling tendencies according to the physics of the solution field (Moin and Yaglom, 1975). A velocity structure function may be expressed as (Babiano et al., 1985; Boffetta and Ecke, 2012; Iyer et al., 2017)

$$S^p(r) = \langle (\boldsymbol{u}(\boldsymbol{x}+\boldsymbol{r}) - \boldsymbol{u}(\boldsymbol{x}))^p \rangle \tag{27}$$

where the ensemble averaging is taken over all positions $\boldsymbol{x}$ and all orientations of $\boldsymbol{r}$ within the computational domain to yield statistics for the length scale $r = |\boldsymbol{r}|$. Our choice of $p$ determines the *order* of the structure function we are examining and this investigation looks at $p = 2$ for the characterization of turbulence in both two and three dimensions.The second-order structure function has been used to charecterize the turbulence both in 2D (e.g., see Babiano et al. (1985)) and 3D (e.g., see Kritsuk et al. (2007)) turbulent flows. We note that some researchers have prefered to use the absolute value definition, which might change the results for odd values of $p$ (e.g., see Arneodo et al. (1996) for a great discussion on various definitions of the structure functions). For the 2D turbulence setting, Babiano et al. (1985) predicted a scaling law of $r^{n-1}$ where $n$ refers to the scaling component of the energy spectrum (i.e., $E(k) \propto k^{-n}$). In 3D turbulence, the scaling of $r^{p/3}$ has been established for the $p$th structure function. Both longitudinal ($\boldsymbol{u} \parallel \boldsymbol{r}$) and transverse ($\boldsymbol{u} \perp \boldsymbol{r}$) third-order velocity structure functions are computed in the present study. In our assessments, a range of $10^{-2} \leq r \leq 10^{-1}$ is assumed to represent the general vicinity of the inertial range.

We utilize the high fidelity data of the previously described numerical experiments for two- and three-dimensional turbulence to obtain structure function statistics at time $t = 5$. Fig. (21) shows the second-order velocity structure function for the longitudinal and transverse directions for the 3D test case. One can observe a steadily increasing alignment with $r^{2/3}$ with decreasing value of $\alpha$ implying weaker compressibility. It is worthy to mention here that Kolmogorov theory dictates a cascade given by $p/3$. Similar trends are observed for both longitudinal and transverse directions suggesting that a certain degree of isotropy now characterizes the system. For ranges of $r$ below $10^{-2}$, it is observed that both longitudinal and transverse structure functions scale according to $r^2$ for the second-order structure function.

We undertake a similar statistical examination for our two-dimensional test case where second-order longitudinal and transverse structure functions are given by Fig. (22) where it is observed that at low $r$, a scaling corresponding to $r^2$ is observed. This is in accordance with findings in Grossmann and Mertens (1992). At larger values of $r$, the $r^2$ scaling transitions to an $r^{4/3}$ scaling at relatively higher compressibility (i.e., $\alpha = 1.0$) and $r$ scaling at $\alpha = 0.25$. Eventually, it is expected that an $r^{2/3}$ behavior must emerge with perfect incompressibility. The aforementioned observations hold true for both longitudinal and transverse second-order structure functions and are consistent with the definition of $S(r) \propto r^{n-1}$. It can be observed that the velocity structure functions for three-dimensional simulations generally obey the prediction of the Kolmogorov theory (for lower values of $\alpha$ indicating weak compressibility) as against their two-dimensional counterparts.

## 4   Conclusions

In this investigation, data from high-fidelity numerical experiments are utilized to study scaling behavior for statistical quantities such as spectra and structure functions. We study two test cases given by the Kelvin-Helmholtz instability problem in two and three dimensions to study spectral scaling laws for compressible shear layer turbulence. Our spectra are given by the averaged kinetic energy magnitude and the averaged density-weighted kinetic energy magnitude and it is observed that while both quantities exhibit similar trends in three dimensions, the density-weighted kinetic energy spectra show varying scaling tendencies in two dimensions. This is demonstrated by a flattening of the density-weighted energy spectra, expected to exhibit $k^{-3}$ scaling in the incompressible limit, to $k^{-7/3}$ scaling for higher compressibility. Variations are also seen in the scaling of the dissipation range. This prompts us to investigate the density power spectrum and the pressure power spectrum for both two and three-dimensional cases and it is observed that two distinct inertial and dissipation range behavior can be observed. For the density power spectrum, both the three-dimensional and two-dimensional cases show a five-thirds scaling behavior in the inertial range with a $k^{-6}$ scaling in the dissipation range. This basically demonstrates that the scaling laws for both kinetic energy and power density spectra coincide with each other only for three-dimensional flows. The pressure power spectrum analysis also demonstrates that the results are less invariant to variations in the compressibility parameter for the two-dimensional KHI problem. The scaling behavior exhibited by the density and pressure power spectra for the two-dimensional test, combined with the trends observed in the energy spectrum and structure function analyses indicate that nonlinear processes exhibiting extreme aspect ratios may have a fundamentally different set of nonlinear interactions as compared to moderate aspect ratios (which may be classified as three-dimensional). Incorporating the effect of boundary conditions, which inevitably leads to large scale

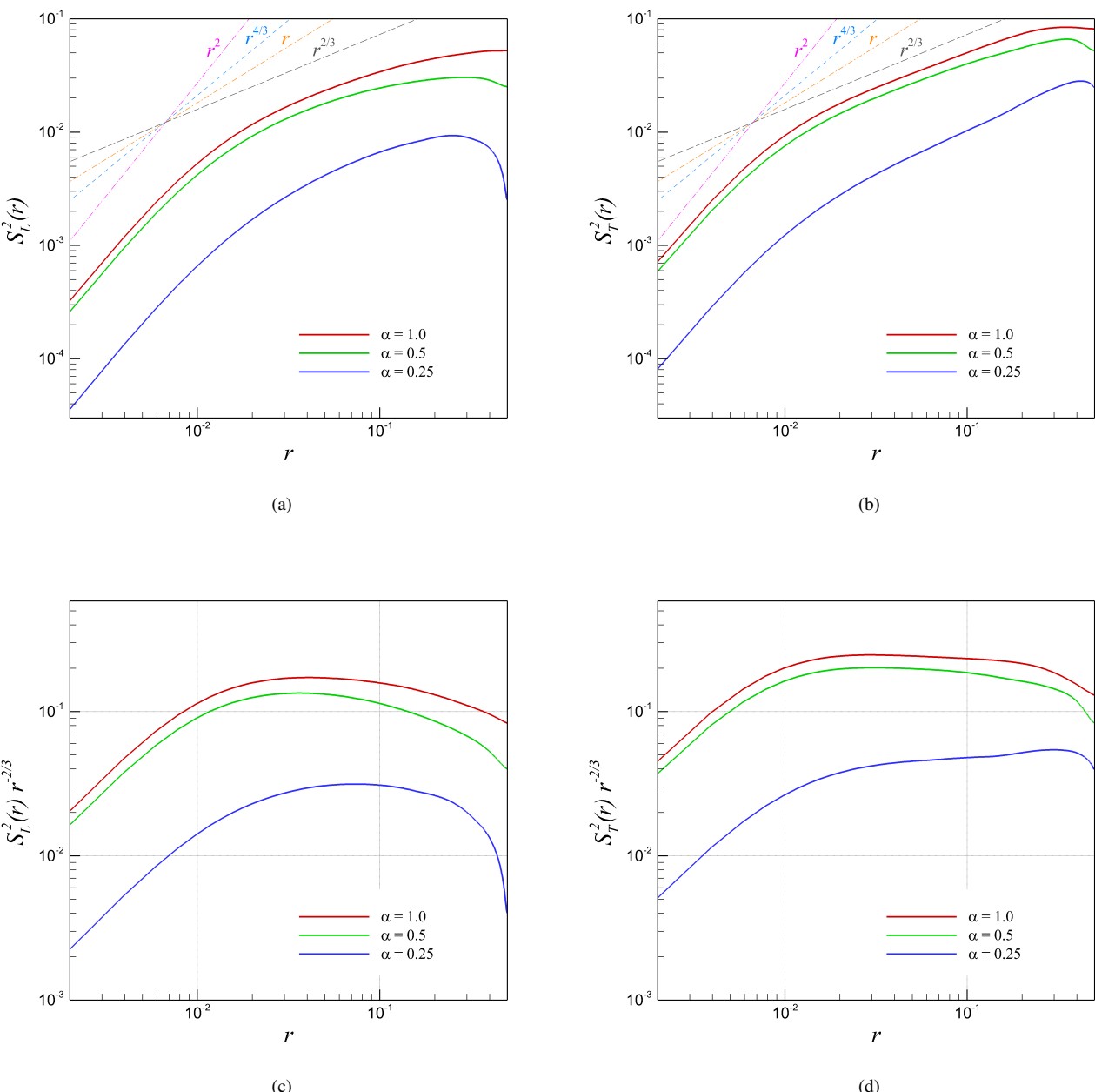

**Figure 21.** Second-order velocity structure functions for 3D KHI turbulence. (a) Longitudinal structure function ($\boldsymbol{u} \parallel \boldsymbol{r}$), (b) transverse structure function ($\boldsymbol{u} \perp \boldsymbol{r}$), (c) compensated form of the longitudinal one, and (d) compensated form of the transverse one.

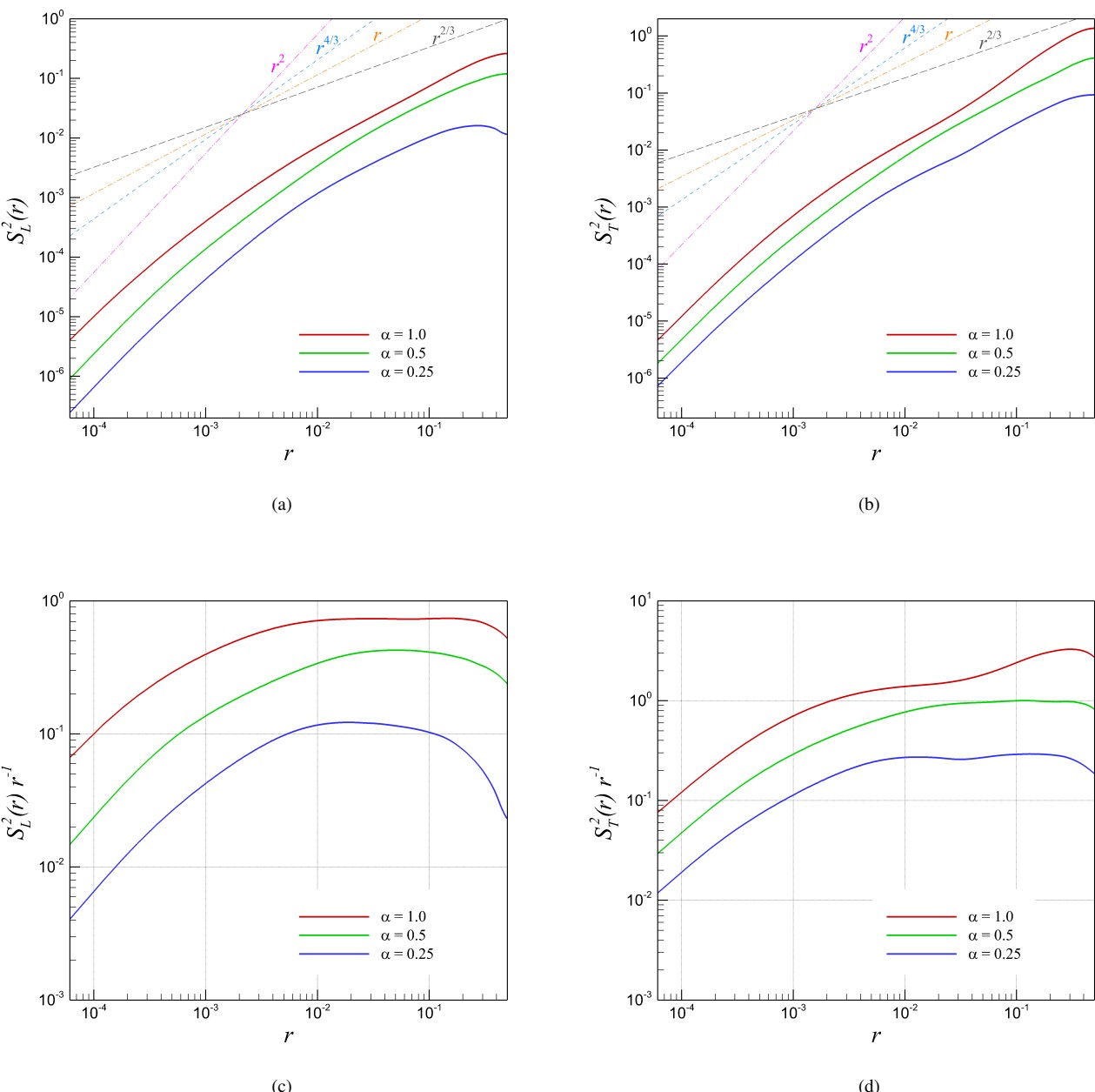

**Figure 22.** Second-order velocity structure functions for 2D KHI turbulence. (a) Longitudinal structure function ($\boldsymbol{u} \parallel \boldsymbol{r}$), (b) transverse structure function ($\boldsymbol{u} \perp \boldsymbol{r}$), (c) compensated form of the longitudinal one, and (d) compensated form of the transverse one.

anisotropy into the scaling tendencies exhibited here would account for further interesting deviations from 3D counterparts. This remains a topic of focus for future investigation.

*Acknowledgements.* The authors are very grateful to the editor and four anonymous referees for their useful comments and suggestions published on the NPGD website that helped us improve the presentation of this paper. The helpful comments from Prof. Bhimsen Shivamoggi (University of Central Florida) and Prof. Bohua Sun (Cape Peninsula University of Technology) are also appreciated. All numerical experiments have been performed using the resources of the Oklahoma State University High Performance Computing (OSU-HPCC) facilities.

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
