# Peer review of "Stratified Kelvin-Helmholtz turbulence of compressible shear flows"

_Nonlinear Processes in Geophysics, 2017_

## Referee Comment (RC1) · Anonymous Referee #1 · 20 Feb 2018

1) The analysis of the flows in the paper is disappointing, since very few of the interesting tools that have been developed for analyzing compressible turbulence in particular have not been used. This includes, but is not limited to:

a) computing energy spectra (and spectral energy fluxes) from the curl-free and divergence-free components of the velocity field via a Helmholtz decomposition, particularly in 2D, since there is a heuristic idea in the literature that compressible 2D turbulence exhibits an inverse-cascade of energy in the divergence-free part of the velocity field while also losing energy to the curl-free part, where it then is transferred to small scales in a direct cascade (mostly via shockwave formation).

b) computing some of the structure functions which have been developed specifically for compressible turbulence, such as that derived and measured in simulations for the 2D case in my own papers https://link.springer.com/article/10.1007/JHEP12(2015)067 (referenced in the paper in question) and https://link.springer.com/article/10.1007/JHEP08(2017)027, or those derived in the 3D case in https://www.cambridge.org/core/journals/journal-of-fluid-mechanics/article/new-relations-for-correlation-functions-in-navierstokes-turbulence/B4E931CFF282345CB20CB3619F3CA27E.

2) The type of velocity structure functions computed in this work (Eq. 20) are not very informative, since the absolute value of the velocity differences is taken inside the average. There are no predictions for the behaviour of such structure functions that do not make ancillary (and strictly invalid) assumptions about the nature of turbulence. On the other hand, the third-order velocity structure function without the absolute value taken indeed has concrete predictions, at least in the incompressible regime (4/5-law in 3D and 3/2-law in 2D). These laws have been extended to the compressible case in the aforementioned papers.

3) On page 12, lines 1-3, a $k^{-3}$ scaling of the energy spectrum is incorrectly stated to be associated with the inverse-energy cascade in Kraichnan-Batchelor-Leith theory of 2D turbulence. It is associated with a direct-cascade of enstrophy in that theory, or an energy condensate otherwise.

4) It is very difficult to compare the various power-law scalings in the plots with the energy spectra or structure functions, since the power-law scalings are placed too far away from those quantities. In my experience, it is very easy to mislead readers with such plots because this style of presenting is very insensitive to errors. The most honest and unforgiving way of presenting that information is to compensate the quantities by the various power laws (i.e. divide the quantites by the expected power laws) and plot on a linear vertical scale.

5) Equation (20) has typos: the argument of the first velocity should be (x+r) and the argument of the second velocity should be (x).

6) On page 11, line 14, a spatial average over the z axis is incorrectly referred to as an ensemble average. Spatial or temporal averages are equivalent to ensemble averages only for ergodic systems, and I believe it is currentlyan open question whether and when turbulence is ergodic.

---

## Short Comment (SC1) · 22 Feb 2018

This artcile study the scaling laws and structure functions of stratified shear flows by performing high-resolution numerical simulations of inviscid compressible turbulence induced by Kelvin-Helmholtz instability. The paper presents scaling laws for some cases, all power exponent of k (wave numver) is within the range of [-2,-5/3]., which is right and have been confirmed before my Bohua Sun [Bohua Sun, Scaling laws of compressible turbulence, Appl. Math. Mech. -Engl. Ed., 2017,38(6): 765–778 (the only HOT paper in the issue)]. Generally speaking, the qulitity of this work is nice and worth to be published.

For your information, enculded please find the copy of my two papers in the field

compressible turbulence.

Please also note the supplement to this comment:
https://www.nonlin-processes-geophys-discuss.net/npg-2017-67/npg-2017-67-SC1-supplement.pdf

**Supplement:**

[supplement omitted: unrelated document]

---

## Author Comment (AC1) · 22 Feb 2018

The authors would like to thank the Reviewer for their time to review our manuscript and to provide valuable comments and suggestions. We view their criticism positively, which we can address when we revise our manuscript. Here we would like to list our preliminary responses to each item raised by the Reviewer:

1-) As discussed in literature (e.g., see recent reviews by Zhou [1,2]), Rayleigh–Taylor and Richtmyer–Meshkov instabilities could introduce modified energy spectra and anisotropy, due to which we believe that some of the discussions there complement our observations in the present manuscript dedicated solely to the Kelvin-Helhmholtz instability (KHI) process. In our manuscript, we have computed two types of energy

spectra for both 2D and 3D KHI flows: (i) from regular velocity components, and (ii) from density weighted velocity components. We note that the way we formulate the problem at hand is special, since both 2D and 3D flows follow the same initial perturbation, which aids us in making an easy comparison of the difference between the 2D and 3D flows and their underlying physics. As highlighted in the manuscript, one of our main discussion points is the difference between density weighing in 2D and 3D flows. In 3D flows, the density spectrum follows the same scaling with the kinematic velocity spectrum, and therefore one may observe more or less similar scaling behavior when spectra are computed from density weighted velocities. In 2D flows, however, the density spectra follows a much shallower (flattened) scaling and this manifests in a substantial difference in the scaling between the spectra obtained from kinematic velocities and density weighted velocities. (a) We believe that the Reviewer's idea on using a Helmholtz decomposition is very useful, which can be done easily in our revision with the use of an FFT. We will follow the suggestion of the Reviewer and split the resolved velocity field into the solenoidal (divergence-free) and compressive (curl-free) components and compute their associated spectra. That will definitely strengthen our discussion. (b) Please see our discussion below in item 2 regarding the computation of the structure functions.

2-) As highlighted by the Reviewer, in the present manuscript, we have used the absolute value of velocity differences when we compute the structure functions. This arises from the definition of the structure function given by Boffetta and Ecke [3] (see their Eq. 4 which uses the absolute value). Aside from Boffetta and Ecke, the absolute value definition has also been used in many other studies (e.g., see [4], Eq. 21 in [5]). On the other hand, many researchers have used the definition without using the absolute value (e.g., [6,7]) and the choice of the computation of the structure function has been discussed by some researchers, for example see [8,9]. In [8], the difference has been identified using Fp (without using the absolute value) and Gp (with using the absolute value). Thanks to the Reviewer, we shall further compare these definitions (with and without the absolute values) when we present the third-order structure functions in our

revision. Indeed, this would be a nice analysis to see the differences in a KHI triggered 2D and 3D flow.

3-) We are in absolute agreement with the Reviewer that the $k^{-3}$ scaling of the energy spectrum is associated with a direct-cascade of enstrophy in KBL theory. This was just a typographic error and we would like to thank the Reviewer for catching it. It will be corrected in our revision.

4-) We understand the Reviewer's concern here. Indeed, that was the reason why we included 4 different straight lines in our plots to give an accurate and fair comparison for each scaling. We believed that we can give the most accurate representation to the readers by that way. We thought that having one compensated line (as suggested by the Reviewer) could be useful but also mislead the reader since there is a slight variation in each case. In our revised manuscript, we can easily compute and present compensated spectra as well. Furthermore, another reason to put 4 systematically scaled lines in each figure (instead of compensating with only one scaling line) is to give a scaling representation beyond the inertial range (i.e., close to the dissipative scales). We noted that although the scaling for inertial range is different in 2D and 3D flows, we can see that the scaling merges to $k^{-6}$ scaling towards the grid cut-off scale for both flows.

5-) The Reviewer is absolutely right. There was a typo in the first term, which should be written as (x+r). We will correct it. Thanks.

6-) This is another place we would like to thank the Reviewer. We had used, unintentionally, the wrong terminology and it should be a "spatial" averaging instead of "ensemble" averaging, and we will correct it when we revise our manuscript.

In conclusion, the authors would like to thank the reviewer again for their time and their valuable comments. Addressing these comments as explained above, we believe that the manuscript can be improved extensively in terms of concept, technical content and clarity of exposition.

Sincerely,

Omer San

References

[1] Zhou, Y. Rayleigh-Taylor and Richtmyer-Meshkov instability induced flow, turbulence, and mixing. I. Physics Reports 723-725, 1-136, 2017.

[2] Zhou, Y. Rayleigh-Taylor and Richtmyer-Meshkov instability induced flow, turbulence, and mixing. II. Physics Reports 723-725, 1-160, 2017.

[3] Boffetta, G. and Ecke, R. E. Two-dimensional turbulence. Annual Review of Fluid Mechanics 44, 427-451, 2012.

[4] Schmidt, W., Federrath, C. and Klessen, R. Is the scaling of supersonic turbulence universal? Physical Review Letters 101, 194505, 2008.

[5] Kritsuk, A.G., Norman, M. L., Padoan, P. and Wagner, R. The statistics of supersonic isothermal turbulence. The Astrophysical Journal 665, 416-431, 2007.

[6] Arenas, A. and Chorin, A. J. On the existence and scaling of structure functions in turbulence according to the data. Proceedings of the National Academy of Sciences of the USA 103, 4352-4355, 2006.

[7] Iyer, K.P., Sreenivasan, K. R. and Yeung, P.K. Reynolds number scaling of velocity increments in isotropic turbulence. Physical Review E 95, 021101, 2017.

[8] Arneodo, A., Baudet, C., Belin, F, Benzi, R., Castaing, B., Chabaud, B., Chavarria, R., Ciliberto, S., Camussi, R., and Chilla, F. Structure functions in turbulence, in various flow configurations, at Reynolds number between 30 and 5000, using extended self-similarity. Europhysics Letters 34, 411-416, 1996.

[9] van de Water, W. and Herweijer J.A. Anomalous scaling and anisotropy in turbulence. Physica B: Condensed Matter 228, 185-191, 1996.

---

## Referee Comment (RC2) · Anonymous Referee #2 · 24 Feb 2018

The authors have studied the spectra of kinetic energy, density-weighted kinetic energy and density, as well as structure functions of velocity in stratified shear flows by performing numerical simulations of inviscid compressible turbulence induced by Kelvin-Helmholtz instability, in both two-dimensional space and three-dimensional space. They have pointed out that there is a significant difference between two spectra of kinetic energy and density-weighted kinetic energy in two-dimensional turbulence due to the effect of density field, while the two spectra of kinetic energy and density-weighted energy in three-dimensional turbulence exhibit the same power-law scaling with the Kolmogorov scaling exponent -5/3. The results are interesting. There are some issues that need to be addressed.

(1) In order to examine the effect of density, two spectra of kinetic energy and densityweighted kinetic energy can be plotted in the same figure. Moreover, authors can plot the spectrum of the difference between the velocity u and the normalized density-weighted velocity \sqrt{\rho}u / <\sqrt{\rho}>, where <\sqrt{\rho}> is the spatial average of square root of density. The effect of density field can be clearly demonstrated by comparing the spectrum of the velocity difference and the spectrum of the velocity itself.

(2) Similar to the previous comments by referee #1, I also suggest that authors should study the spectra of the solenoidal and compressible components of velocity by using the Helmholtz decomposition. The effect of compressibility can be evaluated by the relative magnitude of the compressible component of the velocity spectrum with respect to its solenoidal counterpart.

(3) Please provide some explanations about the -6 scaling of spectra of kinetic energy at high wave numbers (in the dissipation region) since such an exponent of -6 is quite new.

(4) Please cite some recent references about the spectra of velocity and density in two-dimensional and three-dimensional compressible turbulence, and compare the present results with those in the references: (a) http://aip.scitation.org/doi/abs/10.1063/1.4892460 Density distribution in two-dimensional weakly compressible turbulence, D. Terakado and Y. Hattori, Physics of Fluids 26, 085105 (2014). (b) https://journals.aps.org/prfluids/abstract/10.1103/PhysRevFluids.2.092603 How vortices and shocks provide for a flux loop in two-dimensional compressible turbulence, G. Falkovich and A. G. Kritsuk, Physical Review Fluids 2, 092603(R) (2017). (c) https://journals.aps.org/prfluids/abstract/10.1103/PhysRevFluids.2.013403 Spectra and statistics in compressible isotropic turbulence, J. Wang, T. Gotoh, and T. Watanabe, Physical Review Fluids 2, 013403 (2017).

(5) Please plot the root mean square values of velocity and density as functions of time.

Will the results about the spectra od kinetic energy be changed if t=5 is changed to, for example, t=4?

(6) Page 10, line 9: the question mark needs to be deleted.
* * *

---

## Author Comment (AC2) · 26 Feb 2018

The authors would like to thank the Referee for reviewing our manuscript and for providing the authors with their constructive remarks and recommendations, which we have found to be very enlightening. We will address their comments in the revised version of this manuscript. Here is a list of our preliminary responses to their comments:

1-) Thank you for these excellent suggestions. We can perform and post-process the suggested plots to examine the effect of density and these shall be included in our revision.

2-) The authors are in full agreement with both reviewers with respect to this comment on studying the spectra of the solenoidal and compressible components of velocity by

using the Helmholtz decomposition. We will follow this suggestion and incorporate it in our revised form.

3-) We will extend our discussion in the revised text about the -6 scaling towards to the cut-off scale. Regarding this topic, we will also look for additional references.

4-) Thank you for these suggested references, they will be added to the next version of the paper in connection with our results.

5-) During our high resolution simulations, we recorded the field variables at time t=1, 2, 3, 4, and 5 (each single snapshot takes about 25 GB disc space in 2D cases). Thanks to our HPC center facilities, we have been able to store them in our Scratch disk space which is directly connected to our compute nodes (i.e., our data will be readily available to us for further post-processing). Therefore, we can run our post-processing scripts and compute/plot desired spectra at time t=4. Our results indicate that the KHI triggers turbulence before t=1 and the domain was sufficiently well homogenized at t=4 implying that our conclusions would still hold true. We will include our results at different times when we revise our manuscript. Furthermore, during the simulations we have stored time series of the domain integrated total kinetic energy, which we will also include in our revised text.

6-) We thank the Referee for pointing out this typographic error. We shall correct this in the revised manuscript.

We would once again like to thank the reviewer for their time and valuable suggestions, which will undoubtedly lead to a significantly improved manuscript.

Sincerely,

Omer San

---

## Referee Comment (RC3) · Anonymous Referee #3 · 28 Feb 2018

Another interesting quantity to compute and analyze will be the pressure spectrum density. If the full data snapshots have been stored as described in the authors' response to the Referee 2, this can be done without requiring new runs.

Page 3, Line 14-15: might be "through the use of the second- and third-order structure functions with ... "

Page 5, Line 5-6: might be "the utilization of an artificial dissipation in ILES schemes (from ..."

Page 10, Line 9: the reference is missing.

Page 17, Line 17: might be "spectra show" or "spectrum shows"

[Figure]

Page 18, Line 7: the wording "an extremely high aspect ratio fundamentally alters" is unclear and needs revisiting.
* * *

---

## Author Comment (AC3) · 2 Mar 2018

We would like to thank the Referee for reading our manuscript, and for drawing our attention to the scaling of pressure. We are also happy to read that the reviewer finds our manuscript interesting. As discussed in [1], the pressure spectrum can be expressed by $P(k) \sim kE(k)^{2}$. Indeed, this yields the pressure spectra slops of $k^{-7/3}$ for Kolmogorov scaling, and $k^{-5}$ for Kraichnan scaling. We agree that it would be interesting to analyze the behavior of $P(k)$ in our 2D and 3D settings with different compressibilities. We will present our findings related to the scaling of the pressure in our revised manuscript.

We also thank the Referee for suggesting the typographic corrections. We will incor-

porate these in the revision of our manuscript.

Sincerely,

Omer San

[1] Lesieur, Marcel and Ossia, Sepand and Metais, Olivier (1999). Infrared pressure spectra in two-and three-dimensional isotropic incompressible turbulence. Physics of Fluids 11, 1535-1543.
* * *

---

## Referee Comment (RC4) · Anonymous Referee #4 · 6 Apr 2018

Referee's Report on "Stratified Kelvin Helmholtz turbulence of compressible shear "By R. Maulik and O.San

This manuscript uses data from high resolution numerical experiments to investigate scaling behavior of energy – spectra and structure functions in compressible turbulence generated by KHI. The average kinetic energy spectra and the average density weighted kinetic energy spectra are found to exhibit similar trends in 3D but different trends in 2D. This is traced to different scaling behavior shown by the density power spectrum in 3D and 2D. The steepening of the energy spectrum with increasing compressibility as well as the tendency of the structure functions to scale closer to K41 with decreasing compressibility seem to be physically plausible. I recommend publication, subject to several typo corrections and some minor changes,

- Typos: (a) "?" on line 9 on p. 10;
        (b) "E" should "$\hat{E}$" in fig. 8 & 10;
        (c)  eq (20) should be u (x + r) – u (x) ;

- Clean up writing on p. 12 (lines 1- 10) and p. 14 (lines 1-15) and also connect them in a coherent way.

---

## Author Comment (AC4) · 11 Apr 2018

We would like to thank the reviewer for their feedback and we are happy to see their recommendation for the publication of our manuscript. Furthermore, we would like to thank the reviewer once again for bringing our attention to the listed typos and we will fix them when we revise our manuscript.

In conclusion, the authors would like to thank the reviewers again for their time and their valuable comments. Addressing these comments as explained above, we believe that the manuscript can be improved extensively in terms of concept, technical content and clarity of exposition.

Sincerely,

[Figure]

Omer San

---

## Author Response (AR1)

**School of Mechanical and Aerospace Engineering**
218 Engineering North
Stillwater, Oklahoma 74078-5016

Phone: (405) 744-2457
Fax: (405) 744-7873
Email: osan@okstate.edu

May 3, 2018

Professor Ioulia Tchiguirinskaia
Handling Editor
Nonlinear Processes in Geophysics

Dear Prof. Tchiguirinskaia,

Please find attached our revised manuscript entitled *"Stratified Kelvin-Helmholtz turbulence of compressible shear flows"* in accordance with the referees' constructive comments and suggestions, which have helped us construct a stronger manuscript. The authors believe that the manuscript is now significantly improved in terms of concept, demonstration and clarity of exposition.

This letter includes our revisions and our responses to the reviewers. The authors would like to thank the reviewers for their time for reviewing our manuscript and providing their valuable comments. In this letter, we combine and reiterate our previous author response comments available online in the Discussion form. Please find below, a detailed response to each of the comments and recommendations.

**Author Comments to Referee #1:**
The authors would like to thank Reviewer 1 for their time to review our manuscript and to provide valuable comments and suggestions. We have addressed all issues raised in their critique and we believe that our manuscript is now much stronger after addressing these constructive comments. Here we would like to list our preliminary responses to each item raised by the Reviewer:

1-) As discussed in literature (e.g., see recent reviews by Zhou [1,2]), Rayleigh–Taylor and Richtmyer–Meshkov instabilities could introduce modified energy spectra and anisotropy, due to which we believe that some of the discussions there complement our observations in the present manuscript dedicated solely to the Kelvin-Helmholtz instability (KHI) process. In our manuscript, we have computed two types of energy spectra for both 2D and 3D KHI flows: (i) from regular velocity components, and (ii) from density weighted velocity components. We note that the way we formulate the problem at hand is special, since both 2D and 3D flows follow the same initial perturbation, which aids us in making an easy comparison of the difference between the 2D and 3D flows and their underlying physics. As highlighted in the manuscript, one of our main discussion points is the difference between density weighing in 2D and 3D flows. In 3D flows, the density spectrum follows the same scaling with the kinematic velocity spectrum, and therefore one may observe more or less similar scaling behavior when spectra are computed from density weighted velocities. In 2D flows, however, the density spectrum follows a much shallower (flattened) scaling, and this manifests in a substantial difference in the scaling between the spectra

obtained from kinematic velocities and density weighted velocities. (a) We believe that the Reviewers idea on using a Helmholtz decomposition was very useful, which has been done in our revision with the use of an FFT procedure described in the text. We have followed the suggestion of the Reviewer and split the resolved velocity field into the solenoidal (divergence-free) and compressive (curl-free) components and computed their associated spectra. This analysis has been definitely strengthen our discussion. (b) Please see our discussion below in item 2 regarding the computation of the structure functions.

2-) As highlighted by the Reviewer, in the present manuscript, we have used the absolute value of velocity differences when we compute the structure functions. This arises from the definition of the structure function given by Boffetta and Ecke [3] (see their Eq. 4, which uses the absolute value). Aside from Boffetta and Ecke, the absolute value definition has also been used in many other studies (e.g., see [4], Eq. 21 in [5]). On the other hand, many researchers have used the definition without using the absolute value (e.g., [6,7]) and the choice of the computation of the structure function has been discussed by some researchers, for example see [8,9]. In [8], the difference has been identified using Fp (without using the absolute value) and Gp (with using the absolute value). Thanks to the Reviewer, in our revised manuscript we have only presented our results for $p = 2$, which both of the definitions (absolute value or without absolute value) refer to the same expression. Additional references have been also added into our revised text for clarification.

3-) We are in absolute agreement with the Reviewer that the $k^{-3}$ scaling of the energy spectrum is associated with a direct-cascade of enstrophy in KBL theory. This was just a typographic error and we would like to thank the Reviewer for catching it. It has been corrected in our revision.

4-) We understand the Reviewers concern here. Indeed, that was the reason why we included 4 different straight lines in our plots to give an accurate and fair comparison for each scaling. We believed that we can give the most accurate representation to the readers by that way. We also thought that having one compensated line (as suggested by the Reviewer) could be useful but also mislead the reader since there is a slight variation in each case. Furthermore, another reason to put 4 systematically scaled lines in each figure (instead of compensating with only one scaling line) is to give a scaling representation beyond the inertial range (i.e., close to the dissipative scales). We noted that although the scaling for inertial range is different in 2D and 3D flows, we can see that the scaling merges to $k^{-6}$ scaling towards the grid cut-off scale for both flows. In our revised manuscript, following the Reviewer's suggestion, we have computed and presented compensated spectra as well. We agree that our presentation is more clear with these new compensated spectra plots.

5-) The Reviewer is absolutely right. There was a typo in the first term, which should be written as (x+r). We have corrected it. Thanks.

6-) This is another place we would like to thank the Reviewer. We had used, unintentionally, the wrong terminology and it should be a spatial averaging instead of ensemble averaging, and we have corrected it in our revised manuscript.

**Authors Comments to Referee #2:**
The authors would like to thank the Referee for reviewing our manuscript and for providing the authors with their constructive remarks and recommendations, which we have found to be very enlightening. We have addressed all issues raised in their critique and we believe that our manuscript is now much stronger after addressing these constructive comments. Here is a list of our preliminary responses to their comments:

1-) Thank you for these excellent suggestions. We have performed and post-processed the suggested plots to examined the effect of density in our revision.

2-) The authors are in full agreement with both reviewers with respect to this comment on studying the spectra of the solenoidal and compressible components of velocity by using the Helmholtz decomposition. We have followed this suggestion and incorporate it in our revised form.

3-) We have extended our discussion in the revised text about the $k^{-6}$ scaling towards to the cut-off scale. Regarding this topic, we have also added additional references.

4-) Thank you for these suggested references, they have been added to our revised version in connection with our results.

5-) During our high resolution simulations, we recorded the field variables at time t=1, 2, 3, 4, and 5 (each snapshot takes about 25 GB disc space in 2D cases). Thanks to our University's HPC infrastructure, we have our recorded snapshots stored in high fidelity form for any desired post-processing. Therefore, we can run our post-processing scripts and prepare spectrum plots at time t=4. Our results indicate that the KHI triggers turbulence before t=1 and the domain was sufficiently well homogenized at t=4 implying that our conclusions would still hold true. Following the Reviewer, we have added new figures to illustrate the time evolution process. Furthermore, during the simulations we have stored time series of the domain integrated total kinetic energy, which we have also included in our revised text.

6-) We thank the Referee for pointing out this typographic error. We have corrected this in the revised manuscript. We would once again like to thank the reviewer for their time and valuable suggestions, which will undoubtedly lead to a significantly improved manuscript.

**Authors Comments to Referee #3:**
We would like to thank the Referee for reading our manuscript, and for drawing our attention to the scaling of pressure. We are also happy to read that the Reviewer finds our manuscript interesting. As discussed in [10], considering the dimensional

analysis the pressure spectrum can be expressed by $\Pi(k) \sim kE(k)^2$. Indeed this yields the pressure spectra slops of $k^{-7/3}$ scaling for Kolmogorov scaling, and $k^{-5}$ for Kraichnan scaling. We agree that it would be an interesting study to analyze the behavior of $\Pi(k)$ in our 2D and 3D settings with different compressibility ratios. We have presented our findings related to the scaling of the pressure in our revised manuscript. We also thank the Referee for suggesting the typographic corrections. We have been incorporated these in the revision of our manuscript.

**Authors Comments to Referee #4:**
We would like to thank the Reviewer for their feedback and we are happy to see their recommendation for the publication of our manuscript. Furthermore, we would like to thank the Reviewer once again for bringing our attention to the listed typographical issues and we have fixed them in our revised our manuscript.

The authors would like to thank the Reviewer again. We have addressed all these issues in our revised version of the paper.

In conclusion, the authors have revised the existing manuscript in accordance with the reviewers' comments and suggestions. The authors believe that the manuscript is now significantly improved in terms of concept, technical content and clarity of exposition.

Thank you.

Sincerely,

Dr. Omer San

[revised manuscript text omitted]

---

## Author Response (AR2)

**School of Mechanical and Aerospace Engineering**
218 Engineering North
Stillwater, Oklahoma 74078-5016

Phone: (405) 744-2457
Fax: (405) 744-7873
Email: osan@okstate.edu

June 9, 2018

Professor Ioulia Tchiguirinskaia
Handling Editor
Nonlinear Processes in Geophysics

Dear Prof. Tchiguirinskaia,

We would like to thank you for your time in reading our manuscript and your recommendation for the publication of our manuscript in the NPG without the need for further editing. Following your editorial suggestion we have included the Acknowledgments a part of vast recognition to the four anonymous referees.

Please find attached our revised manuscript entitled *"Stratified Kelvin-Helmholtz turbulence of compressible shear flows"* along with our slightly revised Acknowledgment section.

Again, thank you for the reading of our manuscript and providing a valuable feedback.

Thank you.

Sincerely,

Dr. Omer San